# Aromatic nonpolar organogels for efficient and stable perovskite green emitters

Jae-Man Park [1], Jinwoo Park [1], Young-Hoon Kim[1], Huanyu Zhou[1], Younghoon Lee [1], Seung Hyeon Jo[1], Jinwoo Ma[1], Tae-Woo Lee [1,2,3] & Jeong-Yun Sun [1,4 ✉]

Existing gels are mostly polar, whose nature limits their role in soft devices. The intermolecular interactions of nonpolar polymer-liquid system are typically weak, which makes the gel brittle. Here we report highly soft and transparent nonpolar organogels. Even though their elements are only carbon and hydrogen, their elastic modulus, transparency, and stretchability are comparable to common soft hydrogels. A key strategy is introducing aromatic interaction into the polymer-solvent system, resulting in a high swelling ratio that enables efficient plasticization of the polymer networks. As a proof of applicability, soft perovskite nanocomposites are synthesized, where the nonpolar environment of organogels enables stable formation and preservation of highly concentrated perovskite nanocrystals, showing high photoluminescence efficiency (~99.8%) after water-exposure and environmental stabilities against air, water, acid, base, heat, light, and mechanical deformation. Their superb properties enable the demonstration of soft electroluminescent devices that stably emit bright and pure green light under diverse deformations.

[1] Department of Materials Science and Engineering, Seoul National University, Seoul 08826, Republic of Korea. [2] School of Chemical and Biological Engineering, Seoul National University, Seoul 08826, Republic of Korea. [3] Institute of Engineering Research, Research Institute of Advanced Materials, Nano Systems Institute (NSI), BK21 PLUS SNU Materials Division for Educating Creative Global Leaders, Seoul National University, Seoul 08826, Republic of Korea. [4] Research Institute of Advanced Materials (RIAM), Seoul National University, Seoul 08826, Republic of Korea. ✉email: jysun@snu.ac.kr

The rising interest in soft and stretchable optoelectronics has recently harnessed considerable advances in intrinsically stretchable electroluminescent devices (ISELDs)[1–5]. As a potential candidate for next-generation displays, the soft ISELDs should preferably possess high efficiency and narrow emission bandwidth for vivid color information, sufficient stretchability with compliance, and device stability against diverse operating conditions. However, achieving all of the above requirements remains a challenging issue.

Lead halide perovskite nanocrystals (PNCs) have attracted much attention as superb optical sources[6]. In particular, methylammonium lead bromide (MAPbBr$_3$) PNCs are promising green emitters due to their high photoluminescence quantum efficiency (PLQE), narrow full-width at half-maximum (FHWM), low materials cost, and facile synthetic process[7]. However, their practical application is hindered by intrinsic or environmental instability, because low crystal formation energy and the ionic-bonded nature of perovskite structure accelerate degradation when exposed to environmental stresses such as ambient or humid air, water, heat, or light[8].

Strategies of incorporating PNCs into polymer networks have been proposed to enhance external stabilities[9–15]. They are also suitable for preparing solid-state stretchable light-emitting layers, because they can simultaneously provide emitters and matrices. Nevertheless, existing PNC–polymer nanocomposites have suffered from material issues on the proper matrices. Some nanocomposites effectively enhance PLQEs and environmental stabilities, while their rigidity or brittleness is incompatible with stretchable applications[9–11]. While elastomeric nanocomposites have also been reported[12–15], the PNC is physically mixed and dispersed in those matrices; therefore, their luminescence efficiency or environmental stability has remained insufficient for practical application. Furthermore, their elastic moduli are the range of several megapascals, which is incompatible with recently developed soft ISELDs[4]. Therefore, the development of a soft perovskite light-emitting layer that possesses superior properties is still necessitated for the realization of future displays.

Gels composed of rigid polymer network swollen by solvent have recently been utilized as functional components of soft devices[16–18]. While these gels are generally soft, stretchable, and transparent, their role in such devices is mostly limited by their nature; existing gels are mostly made up of polar materials that are capable of hydrogen bonding[16], ion–dipole[17], or dipole–dipole interaction[18], because non-covalently mixed polymer–solvent system requires strong intermolecular forces. The insufficient interaction in gels would hinder swelling, where then polymer chains remain stiff, consequently losing their gel-like mechanical properties.

Some studies have reported swelling behaviors of the organogels, including the system of polystyrene (PS) in toluene[19], or in limonene with an assistance of polyelectrolyte[20], or poly(methyl methacrylate) (PMMA) in toluene or other good solvents[21]. However, the nonpolar property of the organogels, which is an important requisite for the stable formation of PNCs, has little been considered. Therefore, a conception that combines perovskite ionic crystals with nonpolar organogels, which could materialize a mechanically soft perovskite emitter, has also not been studied so far.

Here, we report stretchable and transparent aromatic interaction-induced nonpolar organogels (AINOs) (Fig. 1). The mechanical, optical, and nonpolar properties of the organogels are carefully optimized with the theoretical and experimental considerations of the polymer–solvent system. Taking advantage of the AINO's nonpolar characteristic, luminescent green nanocomposites are synthesized by physically or chemically combining the MAPbBr$_3$ PNCs with the AINOs. The chemically anchored and concentrated nanocomposite is brightly luminescent that it shows considerable luminescence efficiency (up to 99.8%) after water exposure, while possessing high softness and stretchability (Young's modulus of ~100 kPa and stretchability of ~11), whose performance in the form of elastomeric perovskite emitters has not previously been reported to the best of our knowledge. Furthermore, the nanocomposites are highly stable against ambient air, water, acid or base solution, light, heat, and mechanical deformation. To substantiate the applicability of the nanocomposites, fully deformable soft pure green light-emitting devices are herein demonstrated with stretchable alternative current electroluminescent (ACEL) layers as a blue light source, and the nanocomposites as color conversion layers.

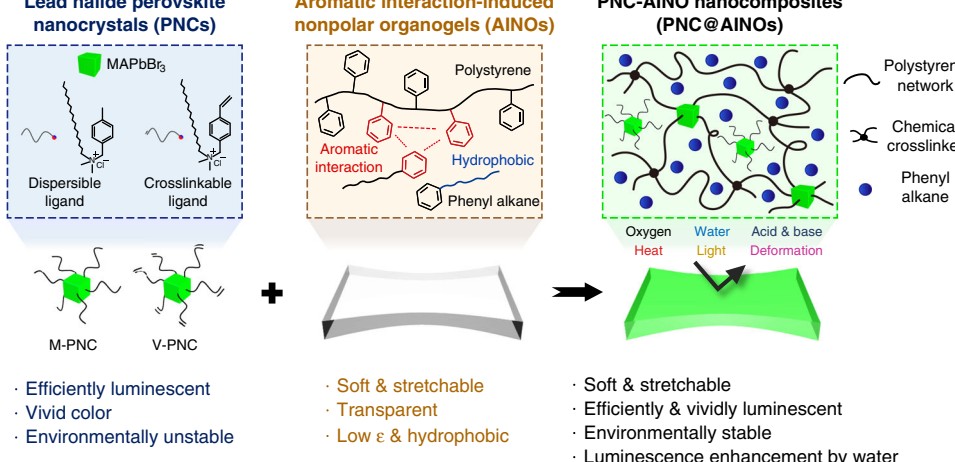

**Fig. 1 Mechanically soft, efficiently luminescent, and environmentally stable perovskite nanocomposites.** Synergistic combination of two materials: (i) methylammonium lead bromide (MAPbBr$_3$) PNCs composed of physically dispersible ligand (M-PNC) or chemically crosslinkable ligand (V-PNC) as highly efficient and pure green emitters, with (ii) aromatic interaction-induced nonpolar organogels (AINOs) consisting of polystyrene network swollen by phenyl alkanes as passivating matrices for the PNCs. Due to the nonpolar and hydrophobic nature of the AINO, the resulting nanocomposites (PNC@AINOs) complement the luminescence and mechanical property of their parents without trade-off, while their nonpolar environment enhances luminescence stability under various environments, such as air (oxygen), water, acid, base, heat, light, and mechanical stress. Furthermore, the luminescence efficiency of the PNC@AINO is increased after water exposure, reaching near unity (~99.8%). ε dielectric constant.

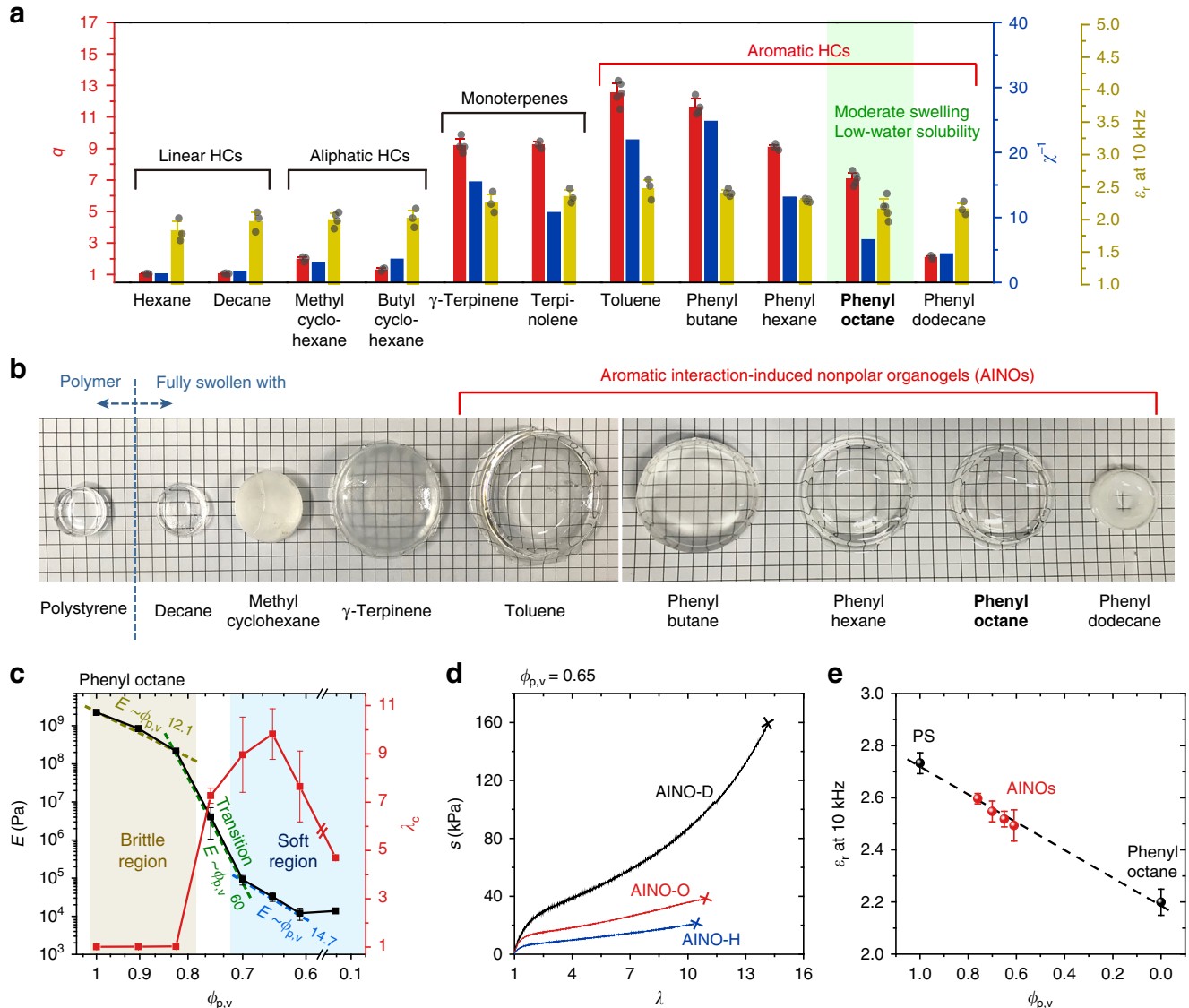

**Fig. 2 Materials investigation for nonpolar organogels. a** Swelling ratio (red columns) and polymer–solvent interaction affinity of various hydrocarbon (HC) solvents with polystyrene (blue columns), and their static dielectric constant at 10 kHz (yellow columns). **b** Photographs of initial polystyrene (PS) and PS organogels fully swollen with various HC solvents. Phenyl octane is chosen as a main solvent component for the AINO, due to its moderate swelling, low dielectric constant, and water solubility. **c** Elastic modulus and rupture stretch of the AINOs based on phenyl octane as a function of their polymer volume concentration from 1 to 0.153 (pure PS to fully swollen state). **d** Stress-stretch curves of the AINOs with different phenyl alkanes at polymer volume concentration of 0.65. **e** Static dielectric constant of the elastomeric AINOs and their constituents (PS and phenyl octane) at 10 kHz. Values in **a**, **c**, and **e** represent the mean and standard deviation (n = 3–5).

## Results

**Aromatic nonpolar organogels**. To investigate the proper components for nonpolar organogels, we consider thermodynamic theories of polymer–solvent system[22]. When polymer and solvent are mixed, the preference of mixing can be explained by introducing polymer–solvent interaction parameter $\chi$ derived from solubility parameter $\delta$ and molar volume of solvent $V_{\mathrm{m,s}}$:

$$\chi = \frac{V_{\mathrm{m,s}}}{RT}\left(\delta_{\mathrm{p}} - \delta_{\mathrm{s}}\right)^2, \qquad (1)$$

where $T$ is the temperature and $R$ is the gas constant (Supplementary Notes 1 and 2 for details). The lower the difference of solubility parameters between polymer $\delta_{\mathrm{p}}$ and solvent $\delta_{\mathrm{s}}$, the more their homogeneous mixing is preferred. We choose PS as a model rigid polymer, because it is a well-known nonpolar

material that is commercially available and facile to synthesize. Then we calculate interaction affinity $\chi^{-1}$ of the polymer with various nonpolar solvents as a parameter for determining the proper solvent components (Fig. 2a). Types of the solvents are confined to hydrocarbons (HCs) to minimize any polar effect from other elements (Supplementary Table 1); therefore, they have static dielectric constants $\varepsilon_{\mathrm{r}}$ below 3 at 10 kHz, representing their high non-polarity (Fig. 2a).

To verify the theoretical consideration of the nonpolar organogel system, we conduct free swelling experiments for the above combinations, which is known to be the empirical approach to determining polymer–solvent interaction[23] (Supplementary Note 3). We first prepare PS polymer networks by free radical polymerization. Their cross-link density is fixed to exclude its effects on swelling behavior. We then measure the swelling

ratio $q$ for comparison with $\chi^{-1}$ and observe their swollen appearance (Fig. 2a, b). Showing a consistent tendency as expected from $\chi^{-1}$ values, linear HCs hardly swell the PS. The PSs swollen by aliphatic HCs become opaque, probably due to phase separation. Monoterpenes moderately swell the PSs but they are translucent. Aromatic HCs effectively swell the PSs from 2 to 12.5 times their initial weight with transparency (except for phenyl dodecane, which has low $\chi^{-1}$ with PS). This result represents that relatively strong non-covalent interaction exists between the PS and the aromatic HCs, although their $\varepsilon_r$ is not much higher than that of the other nonpolar solvents and their constituent elements are only C and H (Supplementary Fig. 1). We suppose it is an aromatic interaction (also known as pi–pi interaction), which comes from structural interaction between aromatic group of the polymers and the solvents[24]. Thus, we name these gels AINOs. We choose phenyl octane as representative solvent for the AINOs, due to its high $q$, wide temperature ranges as liquid state (Supplementary Table 1), and low water solubility (Supplementary Fig. 2e). The same investigations for the PMMA with several nonpolar solvents show similar tendency, but their $q$ is much lower than those of the AINOs, even though their components have much higher values of $\chi^{-1}$ and $\varepsilon_r$ (Supplementary Fig. 3). Since the high swelling ratio of gels ensures their mechanical modulability and softness, the AINO system is selected for further investigations.

Solvent composition in the AINO system significantly affects its mechanical behaviors (Fig. 2c). We conduct mechanical tensile tests for the AINOs synthesized with varying their polymer volume concentration $\phi_{p,v}$ from 1 to 0.61 (Supplementary Figs. 4b and 5). Until $\phi_{p,v}$ reaches 0.824 from 1, the AINOs show hard and brittle behavior, where their elastic moduli $E$ and rupture stretch $\lambda_c$ are from 2.23 GPa to 217 MPa as $\phi_{p,v}$ decreases and range of ~1.03, respectively. On the other hand, when $\phi_{p,v}$ is below 0.7, the AINOs show soft mechanical behavior where their moduli are from 242 to 8.7 kPa as $\phi_{p,v}$ decreases. The $\lambda_c$ increases gradually to 11 at $\phi_{p,v}$ of 0.65, and then decreases as $\phi_{p,v}$ decreases. According to the theory of swollen polymer or rubber elasticity[25], $E$ is exponentially proportional to $\phi_{p,v}$.

$$E \sim \phi_{p,v}^{v}. \tag{2}$$

From previously reported studies with polyacrylamide (PAAm) hydrogel system[26], the exponent $v$ is ~1 for dilute or semi-dilute state ($\phi_{p,v} < 0.7$) and ~45 for concentrated state ($\phi_{p,v} > 0.7$). For the AINOs, $v$ is 12.1 at the brittle region, 14.7 at the soft region, and 60 at the transient region with two transition of power laws across the composition. This high $v$ and two transient regions of the AINOs originate from the relatively low glass transition temperature $T_g$ of the PS (~104 °C), and good interaction of the PS with aromatic HCs; (i) As the composition of solvent in the AINO increases, polymer chains are effectively untangled, so its $T_g$ is decreased rapidly, and (ii) when $T_g$ reaches near room temperature (RT), drastic brittle–ductile transition of its mechanical behavior appears, and finally, (iii) when $T_g$ gets definitely lower than RT, the polymer chains are sufficiently untangled to behave like gels (Supplementary Fig. 6). Therefore, the mechanical behavior of the AINOs can be substantially modulated with a small amount of solvent from brittle but flexible state to ductile elastomeric state, and soft elastomeric state (Supplementary Fig. 7).

The type of solvent also affects the mechanical behavior of the AINOs (Fig. 2d). We conduct tensile tests for the AINOs with a $\phi_{p,v}$ of 0.65, varying the solvent from phenyl hexane (AINO-H) to phenyl octane (AINO-O) and phenyl dodecane (AINO-D). Both $E$ and $\lambda_c$ of the AINOs increase as the number of alkyl chain in phenyl alkane increases, whose values are inversely proportional

to maximum swelling ratio (Supplementary Fig. 8). It indicates that longer phenyl alkanes untangle polymer chains less sufficiently, therefore more stress is needed to elongate them, but more stretch can be undergone.

To investigate the polarity of the AINOs, we conduct dielectric constant measurement. The $\varepsilon_r$ values of AINOs in the soft regions are between 2.48 and 2.6, which is close to the volume-weighted average values of their parents (Fig. 2e and Supplementary Fig. 9). Despite the highly nonpolar environment of the AINOs, their polymer–liquid phases are stably maintained without squeezing out (Supplementary Fig. 10) or separation over a wide range of temperatures from −15 to 50 °C (Supplementary Fig. 11). Reasonably, the AINOs are also intrinsically hydrophobic and highly transparent (Supplementary Figs. 2 and 12), where they could be used as stretchable shielding matrices for the display (Supplementary Movie 1).

**Synergistic combination of the PNC and the AINO.** In comparison with the polar hydrogels[16] or ionogels[17], used as the conductor by hosting dissociated ions, the highly nonpolar environment of the AINOs can harness stable formation and preservation of the ionic crystals. Inspired by the intuitive thought and previous empirical study[27], we incorporate the MAPbBr$_3$ PNCs into the AINOs (Fig. 3a). First, we prepare two types of the PNC reported by Sun et al.[11] in a slightly modified manner, where one type is a chemically crosslinkable PNC (V-PNC) having the vinyl group in its ligands, while the other type is a physically dispersible PNC (M-PNC) as the control group (Supplementary Fig. 4a for synthesis procedure and Supplementary Figs. 13 and 14 for characterizations). The reason for choosing these PNCs is due to their high PLQE (Supplementary Fig. 15) and solubility in the AINO precursors (up to about 12 mg ml$^{-1}$). We then synthesize two types of highly luminescent nanocomposites of the PNCs and the AINOs (M- and V-PNC@AINO) by in situ polymerization (see Methods), where the solvent and $\phi_{p,v}$ of the AINO are chosen to phenyl octane and 0.65, respectively (Supplementary Fig. 4b). The synthesized nanocomposites emit vivid green light under ultraviolet (UV) irradiation (Fig. 3b). Both of the nanocomposites have sharp photoluminescence (PL) spectra with narrow emission bandwidth with no shoulder peaks, exhibiting redshift of the emission peak ($\lambda_{em}$) when undergoing polymerization or increasing concentration of the PNCs (from 518 to 533 nm for the M-PNC, and from 512 to 526 nm for the V-PNC, respectively) (Fig. 3c). The X-ray diffraction results show the clear existence of the PNCs in the AINO, with scattering signals consistent with MAPbBr$_3$ (Supplementary Fig. 16). The PL average lifetime of the PNC@AINOs calculated from the time-resolved PL decay curves are not much different from their solution state, indicating little degradation of the PNCs during the polymerization procedure (Supplementary Fig. 17).

How the PNCs are distributed in the nanocomposites has little effect on their color purity, but greatly affects their luminescence efficiency (Fig. 3d). Both of the nanocomposites have narrow full-widths at half-maximum (FWHMs) in the range 22 ± 1 nm (where ± means variation, except for the M-PNC@AINO with high concentration of the PNC, having ±2 nm variation). The average PLQE values of the M-PNC@AINOs are about 65% regardless of PNC concentration, and reach 72% comparable to the reported nanocomposites of the PNCs and elastomers[13,15], which indicates that the nonpolar environment of the AINOs is compatible to preserve the PNCs. On the other hand, the PLQEs of the V-PNC@AINOs are about 84% on average, and reach 92.1% even for the nanocomposite with saturated concentration (about 12 mg ml$^{-1}$ of V-PNC), which is comparable with

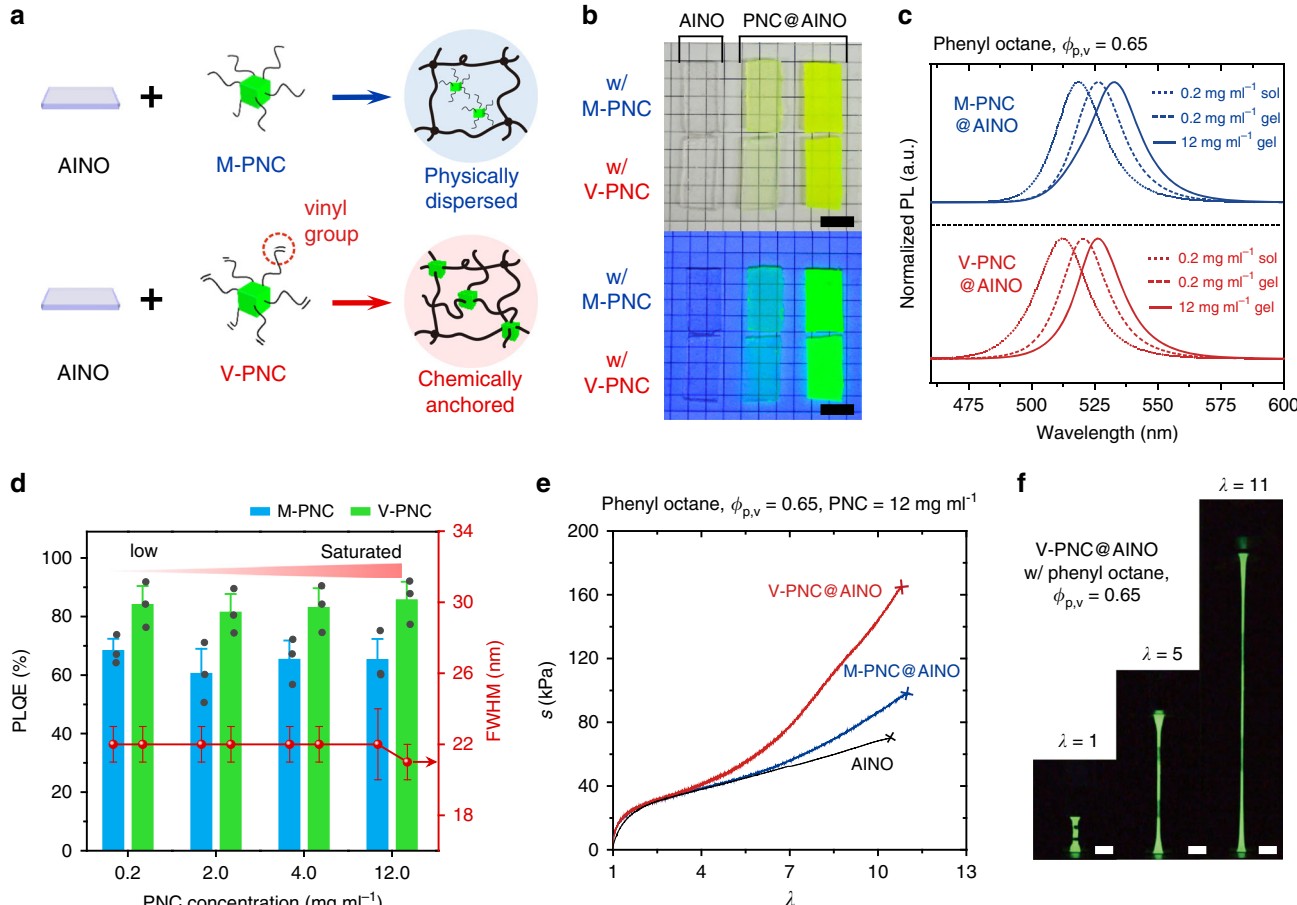

**Fig. 3 Characterizations of PNC@AINOs. a** Schematic illustration of the two kinds of PNC@AINO according to the type of PNC. The M-PNC is physically dispersed in the AINO, while the V-PNC can be chemically anchored to polymer chains of the AINO, due to crosslinkable vinyl groups in its ligands. **b** Photographs of the nanocomposite arrays from the pure AINO (left) to PNC@AINO with low concentration (0.2 mg ml$^{-1}$, middle) and saturated concentration (12 mg ml$^{-1}$, right). First row is the M-PNC@AINO, while the second row is the V-PNC@AINO, under daylight (upper) and UV light (lower). Scale bar = 1 cm. **c** PL spectra of the nanocomposites from precursor state with low concentration (dot line) to its gel state (dash line), and gel state with saturated high concentration (solid line). **d** PLQE and FWHM of the nanocomposites as a function of PNC concentration. Values represent the mean and standard deviation (n = 3). **e** Stress–stretch curves of the AINO, M-PNC@AINO, and V-PNC@AINO. Solvent and polymer volume concentration of each specimen are phenyl octane and 0.65, respectively. The PNC concentration of the M-PNC@AINO and V-PNC@AINO is 12 mg ml$^{-1}$. **f** Photographs of the luminescent V-PNC@AINO stretched to 11 times its original length under UV irradiation. Scale bar = 2 cm.

the PNCs dispersed in toluene or phenyl octane (PLQE ~95%, Supplementary Fig. 15). We suppose that this high PLQE of the V-PNC@AINO comes from homogeneous dispersion of the V-PNCs through the organogel networks, enabled by their ligands that anchor the PNCs chemically to the polymer chains. This anchoring effect prevents their aggregation during polymerization, consequently preventing the self-absorption of PNCs in the synthesized nanocomposite under excitation[28].

The existence of the PNCs in the nanocomposite does not degrade the mechanical properties of the AINOs, but rather makes the nanocomposite more tough (Fig. 3e and Supplementary Movie 2). This toughening behavior is attributed to the role of V-PNC in the nanocomposite; the V-PNC is homogeneously dispersed in the AINO and chemically anchored to polymer chains. That is, the V-PNC behaves as cross-linking point, and therefore it enables the nanocomposite more tough against mechanical stress.

When the nanocomposite is stretched, its PL intensity slightly decreases due to the decrease in the number of emitters in a specific area of the nanocomposites, while their PLQEs are maintained (Supplementary Fig. 18a). Therefore, we can stretch

the V-PNC@AINO to 11 times its original length without rupture, while exhibiting vivid green luminescence under UV irradiation (Fig. 3f). We also demonstrate versatile deformations of the bright green V-PNC@AINO under UV irradiation without luminescent hysteresis (Supplementary Fig. 18b–j).

**Environmental stabilities of the PNC@AINOs.** The V-PNC@AINO is not only efficiently luminescent and highly stretchable but also considerably stable under long-term exposure to various environments (Fig. 4 and Supplementary Figs. 19 and 20). First, the V-PNC@AINO maintains its relative PL intensity, PLQE, and FWHM for more than 100 days in ambient air with moderate blueshift of $\lambda_{em}$ from after 60 days (Fig. 4a and Supplementary Figs. 19a and 20a). The M-PNC@AINO is also somewhat stable for 30 days in air, 10 days in water, and 50 h under UV irradiation (Supplementary Fig. 21).

We observe the clear luminescence increasing behavior of the V-PNC@AINOs during water stability test (Fig. 4b and Supplementary Fig. 19b). For the M-PNC@AINO, the PL intensity and PLQE increase until 24 h after immersion in deionized (DI) water, but then decrease rapidly. On the other

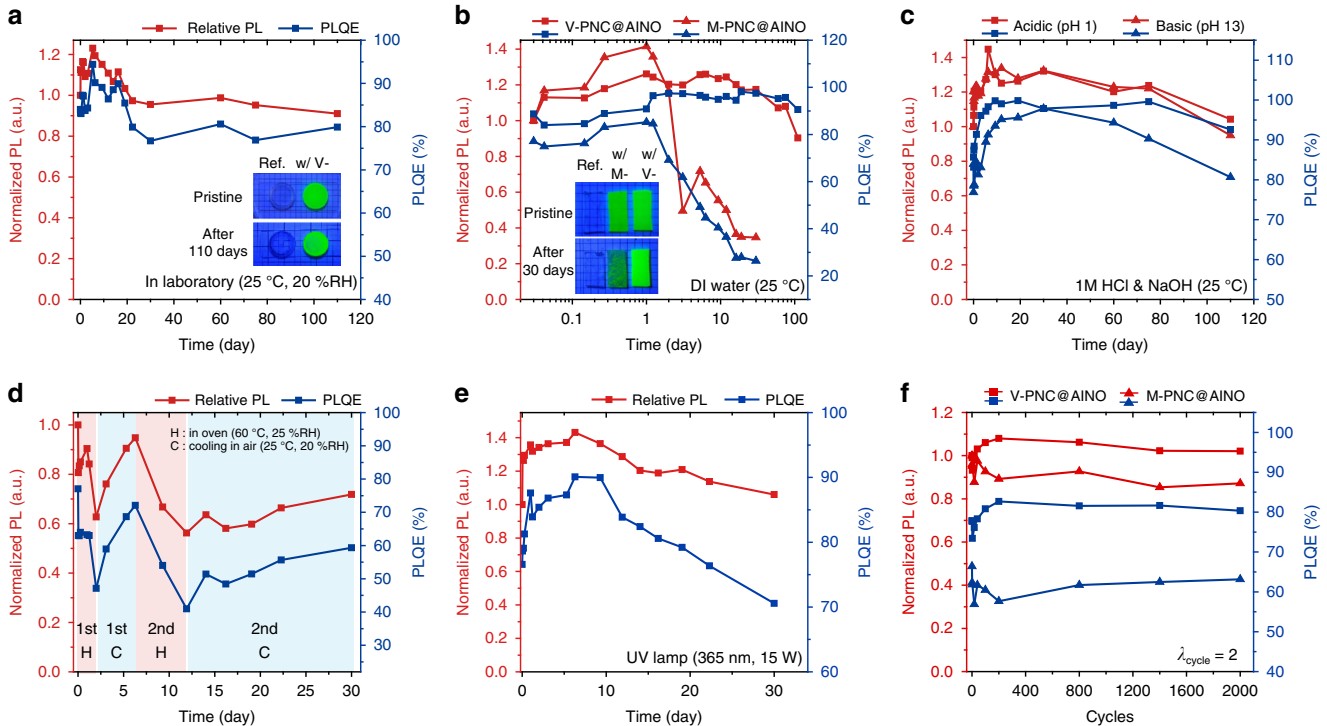

**Fig. 4 Environmental stabilities of the PNC@AINOs. a** Relative PL intensity and PLQE of the V-PNC@AINO in ambient air. Inset shows the reference AINO and the nanocomposite from initial state to after 110 days under air exposure. RH relative humidity. **b** Luminescence behavior of the nanocomposites soaked in DI water clearly differs with the type of PNC. Inset shows the reference AINO (left), M-PNC@AINO (middle), and V-PNC@AINO (right) from initial state to after 30 days in water. **c** The luminescence increasing behavior of the V-PNC@AINO soaked in water is unaffected by its pH. **d** Heat markedly decreases the PL intensity and PLQE of the V-PNC@AINO, but they recover under ambient air. **e** Relative PL intensity and PLQE of the V-PNC@AINO under UV irradiation. **f** Relative PL intensity and PLQE of the nanocomposites during stretch-and-release cycle test. The concentration of the PNC in all nanocomposites for stability tests is 12 mg ml$^{-1}$.

hand, the PL intensity and PLQE of the V-PNC@AINO increase continuously for 2 days after immersion in DI water, where its PLQE reaches >99% with slight blue-shift of $\lambda_{em}$ and broadened FWHM until 60 days (Supplementary Fig. 20b). Also, the nanocomposite retains PL intensity and PLQE compared to its initial values for more than 100 days with additional blue-shift of $\lambda_{em}$ and broadening of FWHM. We propose that this behavior of the V-PNC@AINO in water arises from the following reasons: (i) the elastomeric nature of the AINOs enables the infiltration of water molecules, (ii) the infiltrated water molecules hydrate the surface lead (Pb) of the PNCs, which results in reduction of luminescence quenching site, and consequently enhances PL intensities and PLQE, while (iii) the ligands of the V-PNC, chemically anchored to PS chains, are hard to be dissolved out into water, whereas that of the M-PNC, only physically dispersible in polymer networks, is relatively vulnerable to being dissolved out (Supplementary Figs. 22–25). Release of Pb from the V-PNC@AINO is also delayed by their insoluble ligands (Supplementary Table 2).

This water-induced luminescence increasing behavior of the V-PNC@AINO does not sacrifice its soft mechanical behavior, thus we can still compliantly stretch the V-PNC@AINO taken out from DI water after immersion for 70 days (Supplementary Fig. 26 and Supplementary Movie 3). Furthermore, the pH of water also does not affect the tendency of the behavior that appears in water, reasonably because the nonpolar environment of the AINO hinders the activation of acid or base ions in infiltrated water. Therefore, the V-PNC@AINO have resistance toward acid and base, showing similar luminescence tendency for more than 100 days in acid (pH 1) and base (pH 13) solution (Fig. 4c and Supplementary Figs. 19c and 20c).

Heat certainly decreases the relative PL intensity and PLQE of the V-PNC@AINO to 60% and 45%, respectively, after 50 h in an oven of 60 °C with fluctuation of $\lambda_{em}$ and FWHM (Fig. 4d and Supplementary Figs. 19d and 20d). However, they recover to 90% of their initial values after cooling 100 h in ambient air at RT, which shows similar behavior to the previously reported PNCs[29]. The second cycle of healing and cooling is not as effective as the first cycle.

It is important for the components of ISELDs to be stable under continuous irradiation and mechanical stress. The V-PNC@AINO maintains its PL intensity and PLQE for 30 days under exposure to UV light (UV lamp with power of 15 W and the excited wavelength $\lambda_{ex}$ of 365 nm) with photo-brightening behavior for the first 10 days (Fig. 4e and Supplementary Fig. 19e), while undergoing initial blueshift and decrease of FHWM due to photo-degradation of the PNCs[15] (Supplementary Fig. 20e). In addition, the V-PNC@AINO maintains its PL intensity, PLQE, $\lambda_{em}$, and FHWM after 2000 cycles of 2 times stretch-release (Fig. 4f, Supplementary Fig. 19f, and Supplementary Movie 4), while the M-PNC@AINO shows a slight decrease in PL intensity and PLQE with slight shift or change of $\lambda_{em}$ and FWHM (Supplementary Fig. 20f).

**Fully deformable pure green light-emitting devices.** The outstanding luminescence efficiency, compliant mechanical property, and environmental stability of the V-PNC@AINO enable its practical application to a soft light-emitting layer in recently developed soft ISELD system[4]. We design and fabricate fully deformable soft pure green light-emitting devices (Fig. 5a). We use a typical stretchable ACEL layer based on PAAm hydrogel

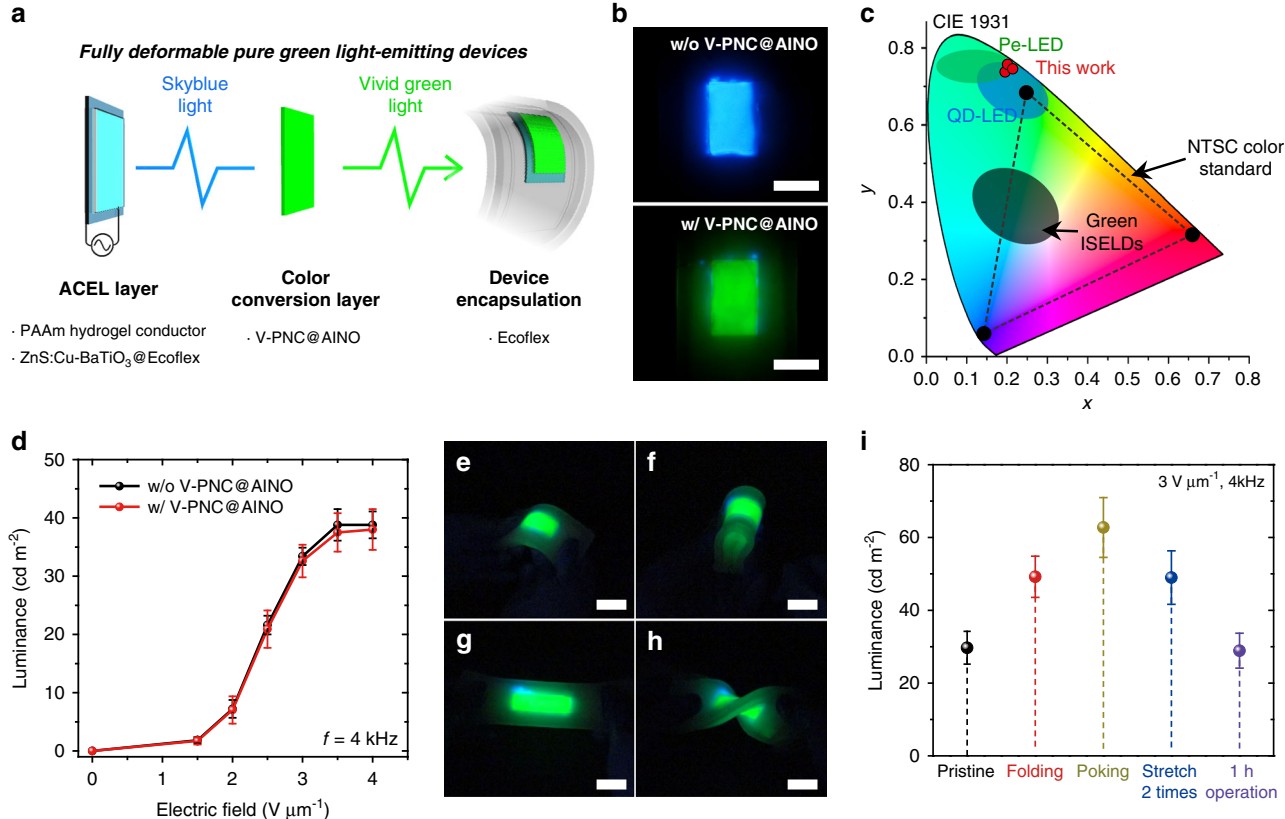

**Fig. 5 Demonstration of fully deformable pure green light-emitting devices. a** Schematic design of the fully deformable pure green light-emitting device, all of whose components consist of soft materials: stretchable alternate current electroluminescence (ACEL) layer composed of hydrogel ionic conductor and elastomeric electroluminescent nanocomposite, V-PNC@AINO as color conversion layer, and entire device encapsulation with silicone elastomer. ZnS: Cu copper doped zinc sulfide, BaTiO₃ barium titanate. **b** Photographs of emission color from the ACEL layer without the V-PNC@AINO layer (upper) and with the layer (lower). Scale bar = 1 cm. **c** Color coordinates of the devices on the Commission Internationale de l'Éclairage (CIE) 1931 color space, plotted together with coordinate ranges of existing green quantum dot-based light-emitting diodes (QD-LEDs) or perovskite LEDs (Pe-LEDs), intrinsically stretchable electroluminescent devices (ISELDs), and National Television System Committee (NTSC) color standard. **d** Luminance of the devices without and with the V-PNC@AINO layer as a function of the applied AC electric field. The frequency of applied voltages is 4 kHz. **e–h** The device emits vivid green light under versatile deformation including bending (**e**), folding (**f**), stretching (**g**), and twisting (**h**). The operating AC voltage and frequency are 2 kV and 4 kHz, respectively. Scale bar = 3 cm. **i** Luminance of the devices under various deformation (folding, poking, and stretching) or after 1 h operation. The operating AC voltage and frequency are 3 kV and 4 kHz, respectively. Values in **d**, **i** represent the mean and standard deviation (n = 3).

conductor, and electroluminescent composites of zinc sulfide doped copper (ZnS:Cu) and barium titanate (BaTiO₃) in Ecoflex 00-30 (ZnS:Cu-BaTiO3@Ecoflex 00-30) as the blue light source. We use the V-PNC@AINO with PNC concentration of 12 mg ml⁻¹ as the color conversion layer after soaked in water for 7 days, because the water-soaked nanocomposite has higher absorbance in the visible light range, which makes the color-conversion more efficiently (Supplementary Note 4 and Supplementary Figs. 27 and 28). The interfaces between each pair of adjacent layers are bonded with diluted cyanoacrylate adhesives to prevent delamination under mechanical stress[30], and all components are encapsulated with Ecoflex 00-30 (Supplementary Fig. 29). The frequency of AC voltages applied is fixed at 4 kHz (Supplementary Fig. 30).

The operating principle of the devices is the same as using phosphor for color conversion in liquid crystal display television[6]; when an AC electric field is applied between the conductors, the ACEL device emits bright sky blue light, and excites the V-PNC@AINO to emit its vivid green light (Fig. 5b). Consequently, the devices have high-purity green color coordinates of (0.176, 0.780), (0.182, 0.750), or (0.201, 0.761) on Commission Internationale de l'Éclairage (CIE) 1931 color space (Supplementary

Fig. 31a). Those values are beyond National television system committee (NTSC) color standard and close to those of quantum dot-based light-emitting diode (QD-LED) or perovskite-based LEDs (Pe-LEDs), for the first time in the form of the soft ISELDs (see Fig. 5c and Supplementary Fig. 31c for the detail color coordinates).

Since the V-PNC@AINO is also efficiently luminescent under excitation of the blue light series (Supplementary Fig. 31b), the devices little lose their luminance in color-conversion process (Fig. 5d). With increasing electric field, the luminance also increases for both of the devices with and without the color conversion layer. The color conversion efficiency of the device is about 97% regardless of the applied electric field (Supplementary Note 5 and Supplementary Fig. 32). Since the intensity of the applied electric field barely affects the color purity, the brightness of the devices can be modulated without change of color information (Supplementary Fig. 33). The power conversion efficiency of the device is about 413.6 mlm W⁻¹ (Supplementary Note 6 and Supplementary Fig. 34).

Since all the constituents of the device are soft, it has the elastic modulus of ~52 kPa (Supplementary Fig. 35a). Therefore, we demonstrate a manual operation of the device, emitting bright

and vivid green light under diverse deformations such as bending, folding, stretching, and twisting without device failure, delamination, or luminescent hysteresis (Fig. 5e–h and Supplementary Movie 5). Furthermore, the device maintains its luminance and color purity for 1 h in operation, cyclic stretch-and-release operation, or operation with time interval (Supplementary Figs. 31c, 36).

The luminance of the devices increases during deformation (Fig. 5i and Supplementary Fig. 35b), reasonably because the thickness of ACEL layer is reduced under deformation, and therefore the electric field applied to the layer is increased at the same applied voltages. The color coordinate of green light emitted from the devices is slightly changed under deformation (Supplementary Figs. 31c and 35c), but still remains pure, exceeding the NTSC color standard. The color purity of the devices will be improved by reducing the transmission of excitation light. It can be achieved with more densely dispersed PNC-organogel nanocomposites, enabled by introducing more soluble PNCs to the organogel precursors, such as a kind of PNCs having longer or branched capping ligands[31].

## Discussion

In this study, we report a type of stretchable emitter consisting of PNCs and nonpolar organogels based on aromatic interaction. PS, as an aromatic polymer network, is efficiently plasticized by the aromatic HCs. Its elastic modulus decreases from gigapascals to kilopascals by just 0.3 volume ratio of the solvent, which endows widely tunable mechanical behavior of the nonpolar organogels. The aromatic nonpolar organogels can also serve as transparent scaffolds or dielectric elastomers in soft devices. Furthermore, the underlying theories for characterizing the organogels consistently accord with the experimental results, which could provide a general guideline for engineering diverse functional polymer networks with liquids.

The synergistic strategy of incorporating PNCs chemically into the highly nonpolar organogels brings improved luminescence properties and stabilities toward various environmental conditions, with finding that the water can enhance the luminescence efficiency of concentrated PNC in nonpolar elastomers. The PLQE of near unity is achieved in the form of elastomeric perovskite emitters. With consideration of ligand and compositional engineering[32], diverse perovskite composites with the nonpolar organogels can be also presented as soft color conversion layers having wide color gamut, tunable mechanical property, and external stability.

Our work of demonstrating soft pure green electroluminescent devices does not address the issues that existing ISELDs have, such as the high-voltage need for operation, low luminescence intensity, or low resolution and density of pixels, but rather focuses on the color purity, which can provide a fundamental requisite for upcoming deformable displays. The perovskite-aromatic nonpolar organogel nanocomposites may enable a step forward in soft optoelectronic devices.

## Methods

**Materials.** Styrene, methyl methacrylate (MMA), divinylbenzene (DVB), ethylene glycol dimethacrylate (EGDMA), toluene, dimethylformamide (DMF), ethanol, acetone, n-butanol, hexane, decane, 1,4-dioxane, methyl cyclohexane, γ-terpinene, terpinolene, anisole, 4-vinylbenzyl chloride, 4-methylbenzyl chloride, lead bromide (PbBr₂), methylammonium bromide (MABr), acrylamide (AAm), N,N-methylene-bisacrylamide (MBAAm), lithium phenyl-2,4,6-trimethylbenzoyl-phosphinate (LAP), lithium chloride (LiCl), barium titanate (BaTiO₃), and 2,2,4-trimethylpentane (TMP) were purchased from Sigma-Aldrich. Phenyl alkanes, amyl acetate, butyl cyclohexane, and n,n-dimehtyl-octadecylamine were purchased from TCI chemicals. 2,2-Azobisi-sobutyronitrile (AIBN), hydrochloric acid (HCl, 35% solution), and sodium hydroxide (NaOH) were purchased from Daejung Industry Co., Ltd. Zinc sulfide doped with copper (ZnS:Cu) phosphor was purchased from Lonco Company Ltd, while Ecoflex 00-30 was purchased from Smooth-On Inc.

Styrene and MMA monomers were distilled, while DVB and EGDMA chemical crosslinkers were passed through basic alumina column before use. An AIBN radical initiator was recrystallized from ethanol. Phenyl alkanes were dried using 4 Å molecular sieves, and other chemicals were used as received.

**Preparation of PS and PMMA networks for swelling test.** To synthesize the PS, we dissolved DVB of 0.086 mol% and AIBN of 0.074 mol% in styrene. The solution was degassed with freeze–pump–thaw cycles at least one time to remove dissolved oxygen. Subsequently, the degassed solution was moved to the glove box, poured into the glass vials with fixed volume, and polymerized at 65 °C for 24 h. Disc-shaped PSs of 2 cm diameter and 5 mm height were obtained after removing glass vials. The same procedure was repeated for preparing disc-shaped PMMA, while chemical crosslinker is replaced to EGDMA with the same molar ratio as for DVB.

Then, we swelled the PS and PMMA samples by soaking them in various nonpolar solvents for 7 days at 25 °C, exchanging solvents every 3 days. We measured swelling ratio q, as defined by the following equation:

$$q = \frac{W_{\mathrm{f}} - W_{\mathrm{i}}}{W_{\mathrm{i}}}, \tag{3}$$

where $W_{\mathrm{i}}$ and $W_{\mathrm{f}}$ are the weights of the initially dried polymers and fully swollen organogels, respectively.

**Preparation of the AINOs.** We dissolved 0.086 mol% DVB and 0.074 mol% AIBN in styrene. Then, we added the desired amounts of phenyl alkanes into the solution where the volume concentration of the solvents varied as (0.096, 0.176, 0.237, 0.3, 0.349, and 0.391) (these values are matched with $\phi_{\mathrm{p,v}}$ of (0.904, 0.824, 0.763, 0.7, 0.651, and 0.609), respectively). After freeze–pump–thaw cycles of at least one time, the solutions were moved into glove box, poured onto glass mold with anodized aluminum oxide spacers, and polymerized at 75 °C for 100 h. After being carefully detached from glass mold, the transparent AINOs with various compositions were obtained. We did not prepare the AINOs with volume concentration of solvents more than 0.391 (lower than $\phi_{\mathrm{p,v}}$ of 0.609), because of the too lengthy time required for complete polymerization due to their low monomer concentrations. To prepare fully swollen AINO with the volume concentration of phenyl octane of 0.847 ($\phi_{\mathrm{p,v}}$ of 0.153), we freely swelled the AINO with phenyl octane prepared by the above procedure.

**Solvent composition and glass transition temperature of the AINOs.** To verify the solvent content in the AINOs, thermogravimetric analysis (TGA) was conducted with a temperature range of 50–600 °C at 10 °C min⁻¹ (Discovery TGA, TA Instrument). The $\phi_{\mathrm{p,v}}$ of the AINOs was calculated by the following equation:

$$\phi_{\mathrm{p,v}} = \frac{V_{\mathrm{p}}}{V_{\mathrm{p}} + V_{\mathrm{s}}} = \frac{\rho_{\mathrm{p}} m_{\mathrm{s}}}{\rho_{\mathrm{p}} m_{\mathrm{s}} + \rho_{\mathrm{s}} m_{\mathrm{p}}}, \tag{4}$$

where $V_{\mathrm{s}}$ and $V_{\mathrm{p}}$ are the volume of the solvent and polymer, $m_{\mathrm{s}}$ and $m_{\mathrm{p}}$ are the relative weights of the solvent and the polymer, while $\rho_{\mathrm{s}}$ and $\rho_{\mathrm{p}}$ are the density of the solvent and the PS, respectively ($\rho_{\mathrm{s}} = 0.86\ \mathrm{g\,cm}^{-3}$ for phenyl octane, $\rho_{\mathrm{ps}} = 1.02\ \mathrm{g\,cm}^{-3}$ for PS).

To verify the glass transition temperature ($T_{\mathrm{g}}$) of the AINOs according to their composition, we conducted the differential scanning calorimetry (DSC) experiment in a heat–cool cycle (Discovery DSC, TA Instrument). The sample of 5–10 mg was placed in a non-hermetic pan, and an empty pan was used as a reference. All of the samples were firstly heated from −50 to 120 °C with 5 °C min⁻¹, jumped to −50 °C with 10 °C min⁻¹, and secondly heated from −50 to 120 °C with 5 °C min⁻¹, wherein the thermal transitions for the heating cycle were recorded. The $T_{\mathrm{g}}$ was determined by the inflection point of the heat capacity in the second cycle.

**Synthesis of capping ligands: VBDSAC and MBDSAC.** To prepare the ligands for synthesizing PNCs: 4-vinylbenzyl-dimethylstearyl ammonium chloride (VBDSAC) and 4-methylbenzyl-dimethylstearyl ammonium chloride (MBDSAC), we followed a method of previous study with slightly modified manner[11]. Briefly, 4.2 ml of N,N-dimethyloctadecylamine was dissolved in 3 ml of acetone, and 9 ml of 4-vinylbenzyl chloride for chemically crosslinkable ligand (VBDSAC) or 8.45 ml of 4-methylbenzyl chloride for physically dispersed ligand (MBDSAC) was added dropwise to the solution at 25 °C. Then, the mixtures were heated at 40 °C for 2 h under stirring. After cooling, the mixtures were washed with acetone three times, followed by filtration, and finally, dried under vacuum overnight at 25 °C.

**Synthesis of the PNCs: V-PNC and M-PNC.** The PNCs: Chemically crosslinkable PNC (V-PNC) and physically dispersed PNC (M-PNC) were synthesized following the previously reported method with slightly modified manner[11]. The 0.066 g of MABr, 0.367 g of PbBr₂, 0.18 g of VBDSAC for V-PNC or 0.176 g of MBDSAC for M-PNC were dissolved in 1 ml of DMF under stirring at 80 °C, respectively. The three clearly dissolved solutions were cooled and mixed together, and subsequently added into 60 ml of toluene under vigorous stirring. The obtained yellow-green PNC solutions were centrifuged at 4000 r.p.m. for 5 min, and the supernatants were collected and washed with n-butanol two times to remove the excess ligands where

the volume ratio of toluene to *n*-butanol is 1:2. Finally, the precipitants were dried under vacuum at 25 °C overnight before further use.

**Synthesis of the PNC@AINO nanocomposites**. The nanocomposite of the PNC and the AINO was prepared by in situ polymerization. We first prepared the AINO solutions by dissolving 0.086 mol% of DVB and 34.9 vol% of phenyl octane into styrene. The V-PNCs or M-PNCs were dissolved in the AINO solutions with desired concentration under stirring, from low concentration (about 0.2 mg ml$^{-1}$, centrifuged after dissolving) to saturated state (about 12 mg ml$^{-1}$). Then the AIBN was placed in a Schlenk reaction glass tube, and the atmosphere changed to inert state by vacuum-N$_2$ gas purging cycles three times. The PNC@AINO solutions were injected and sonicated for 5 min to dissolve AIBN into the solutions. After freeze–pump–thaw cycles of at least three times, the purified solutions were moved to the glove box and poured onto glass mold with anodized aluminum oxide spacers whose thickness were 1 or 2 mm, followed by polymerization at 65 °C for 100 h.

To compare the performance as perovskite nanocomposite emitters, we fabricated the nanocomposite based on the V-PNC and the poly(dimethylsiloxane) (PDMS, Sylgard 184, Dow Corning), which is similar siloxane-based polymer with Ecoflex 00-30. We mixed 10 ml of the V-PNC solution (~1 mg ml$^{-1}$ in toluene) with 10 g of PDMS mixture (10:1 weight ratio of precursor and curing agent), dried toluene under vacuum, and poured onto glass mold with 1 mm thickness. After curing overnight at 60 °C in the glove box, the V-PNC and PDMS nanocomposite (V-PNC@PDMS) sample was obtained.

**Characterization of the PNCs and the PNC@AINO nanocomposites**. For characterization of the synthesized surfactant ligands for the PNCs, we conducted nuclear magnetic resonance (NMR) measurements. The $^1$H-NMR spectra were obtained by a 400 MHz NMR spectrometer (JeolJNM-LA400 with LFG, JEOL). Chloroform-d (CDCl3, δ 7.26 ppm) was used to calibrate chemical shifts. For characterization of the synthesized PNCs, transmission electron microscopy images at 200 kV (Tecnai F20, FEI), X-ray photoelectron spectra analysis data (AXIS-HSI, KRATOS), and X-ray diffraction spectra analysis data from 5° to 55° (New D8 Advance, Bruker) were collected, respectively. PL spectra of the PNCs and the PNC@AINOs were measured by a spectrofluorometer (JASCO FP-8500). Photo-luminescence quantum yield (PLQE) for the PNCs was measured by dissolving the PNCs in styrene-phenyl octane solution whose volume concentration of the styrene was 0.65, and pure solution was used as reference using the same spectro-fluorometer with 100 nm integrating sphere (ILF-835) and Jasco SpectraManager II Software. PLQE of the nanocomposites was measured using the same procedure, while the pure AINOs of the same thickness and size were used as references. FWHMs of the PNCs and the nanocomposites were calculated from the raw data of their PL spectra. The excitation wavelength for all PL spectra was fixed to 365 nm, except for obtaining PL spectra and PLQEs excited by blue light sources. Absor-bance spectra were obtained by a UV-Vis spectrometer (Cary 60 UV-Vis, Agilent Technologies). Transmittance of the AINOs in the range 300–800 nm was obtained by the same spectrometer. References were air and glass cuvette filled with DI water for the film state of AINO and the liquid state of phenyl octane, respectively. Time-correlated single photon counting (TCSPC) measurements were conducted by a picosecond-pulse laser head (LDH-P-C-405B, PicoQuant) with 405 nm excitation wavelength and 40 MHz repetition rate was used as an excitation source. The emitted light was resolved and detected by a monochromator (SP-2155, Acton) and TCSPC module (PicoHarp, PicoQuant) with MCP-PMT (R3809U-50, Hama-matsu), respectively. Fourier transform-infrared (FT-IR) spectra analysis data of the AINO and nanocomposites were collected by an FT-IR spectrometer (Nicolet iS 10, Thermo Fischer Scientific). All photographs of the luminescent PNC@AINO nanocomposites under UV irradiation were filmed by a DSLR camera with fixed aperture—f/2.8 and ISO-800 for visual comparison.

**Mechanical tests**. To minimize slippage during tensile test, all specimens of the AINOs and the PNC@AINO nanocomposites were cut into a dumbbell shape with standard DIN 53504-type S3a (50 mm in overall length, 8.5 mm in shoulder width, 4 mm in narrow width, 2 mm in thickness and 10 mm in gauge length) using a homemade metallic stamp for soft specimens and a laser cutter (Universal Laser System, VLS 3.50) for hard specimens. The specimens were mounted into the holding grips of a universal testing machine with a 50 N capacity load cell for soft specimens (Instron 3343) and 100 kN capacity load cell for rigid specimens (Instron 5582). To prevent slippage between the jaw faces and soft specimens, 60 grit silicon carbide paper (Allied High Tech Products, Inc.) was bonded with superglue to jaw faces in the tensile machine. For compression test, the specimens were fabricated to disc shapes of 20 mm diameter and 130 mm height, and com-pressed to 30% of their initial heights. All mechanical tests, including tensile test, compression test, and loading–unloading cycle test, were performed at 25 °C. Stretch rates were kept constant at 6 min$^{-1}$ for soft specimens and 2 min$^{-1}$ for hard specimens, and strain rate was kept constant at 0.5 min$^{-1}$ for compressive tests.

**Dielectric constant measurement**. To investigate the polarity of the solvents, polymers, and organogels, we conducted dielectric constant measurement using

LCR meter (E4980A precision LCR meter, Agilent Technologies) equipped with accessories of dielectric test fixture and software (16451B and N1500A). Frequency sweeps were performed from 200 Hz to 1 MHz. Dielectric constant of the solvents was measured by non-contact method, while that of AINOs was measured by contact method with 5 mm diameter active electrode and 1 mm height to obtain their capacitance.

**Hydrophobicity test**. To investigate the hydrophobicity of phenyl alkanes, PS, and the AINOs, we measured water solubility of phenyl alkanes for the liquids and contact angles of water micro-droplets on the solid specimens, respectively. Water solubility of the phenyl alkanes was measured using the Karl–Fischer titration method (MKV-710M, KEM). The contact angles between 5 μl droplets of DI water and the solid samples were measured using a contact angle analyzer (FEMTOFAB, Smart Drop) at 25 °C. The mean values were measured three times, and different positions of the samples were used.

**Volatility test**. To investigate the volatility of the AINOs, we performed eva-poration test by measuring relative weight of the samples whose initial weights were about 3 g by time in laboratory at atmospheric pressure.

**Environmental stability test**. The environmental stabilities of the PNC@AINO nanocomposites were investigated by measuring the relative PL intensity, PLQE, PL emission peak, and FWHM of the specimens from their PL spectra, with photographs of the samples that were under daylight or UV irradiation as visual reference. For heat stability test, we placed phenyl octane in vials near the samples to minimize their evaporation issues. For the DI water, acid, and base stability test, we placed the samples (about 1 g) in each liquid or solution (100 ml) and changed those liquids every 2 days. The pH of acid (HCl) and base (NaOH) solution was fixed to 1 and 13, respectively, by 1 M solution titrated with a pH meter (Seven2Go pH/mV meter, Metller-Toledo). For the mechanical stability test, the reference AINOs and the nanocomposites were stretched together using the universal testing machine (Instron 3343). All of the PL spectra and PLQE values for the stability tests were obtained by measuring the same locations of the samples as the excited region. All the reference photographs of luminescent nanocomposites under UV light were taken by a DSLR camera, fixed aperture—f/2.8 and ISO-800 for visual comparison.

To investigate the environmental toxicity, we measured the release content of Pb during water soaking test of the nanocomposites by investigating the water used for soaking the samples using inductively coupled plasma-atomic emission spectroscopy (ICP-730ES, Varian).

**Fabrication of fully deformable green-light-emitting devices**. To synthesize hydrogel conductors, we dissolve AAm and LiCl in DI water at 2.2 and 4 M, respectively. The 0.06 wt% of MBAAm and 0.17 wt% of LAP to AAm were also added, and degassed in a vacuum chamber. The solutions were poured into 1-mm-thick acryl mold, and cured using a UV light crosslinker (UVC 500, Hoefer) for 20 min with a power of 8 W and a wavelength of 365 nm.

To fabricate the electroluminescent layers, we mixed ZnS:Cu and BaTiO$_3$ powder with Ecoflex 00-30 (1:1 weight ratio of precursor A and B) with weight ratios of 3:2:6, followed by degassing under vacuum. The mixtures were poured into 1-mm-thick acryl mold, and cured in an oven at 80 °C for 30 min.

To fabricate the encapsulating matrix, we mixed Ecoflex 00-30 precursors with a weight ratio of 1:1, and degassed in a vacuum chamber. The mixtures were poured into embossed acryl mold, and cured in an oven at 80 °C for 30 min.

To synthesize bridge hydrogel for adhesives of the V-PNC@AINO and Ecoflex 00-30, we used the same recipe and procedure for preparing hydrogel conductors, except for excluding LiCl, and changing the thickness of mold to 300 μm.

To assemble the entire components, we dissolved 20 vol% cyanoacrylate adhesives (Loctite 406, Henkel Adhesives) in TMP. Before the deposition of each layer (between hydrogels and Ecoflex 00-30 for electroluminescence or encapsulation and the V-PNC@AINO), the solutions were spin-coated at 2000 r.p.m. for 30 s in Ecoflex 00-30, and the V-PNC@AINO, followed by careful attachment with hydrogels. Copper wires were inserted through encapsulating Ecoflex 00-30 to the hydrogel conductors.

**Operation and characterization of the fully deformable green-light-emitting devices**. To operate the devices, AC voltage was generated using a high-voltage amplifier (TREK) and a function generator (Agilent Technology).

To measure the luminance and color coordinate of the devices under operation, a luminance and color meter (CS-150 with close-up lens, Konica Minolta) was located 20 cm in front of the devices, and the luminance and color coordinate of the devices were measured under varying applied voltages, frequency, and versatile deformation.

To measure the power consumption of the device, we observed the current by connecting the device with an oscilloscope (DPO5104B, Tektronix) and a low-noise current preamplifier (Stanford research systems, SR570) in series, while applying AC voltages of 3 kV with 4 kHz.

## Data availability
The data that support the findings of this study are available from the corresponding author upon reasonable request.

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

## Acknowledgements
The authors thank Y. H. Cho in Seoul National University (SNU) Materials Science and Engineering Department for his help with mechanical tensile test for hard specimens. This work was supported by LG Display under LGD-SNU Incubation Program. This work was supported by a National Research Foundation of Korea (NRF) grant funded by the Korean Government (No. 2017M3D9A1073922 and NRF-2018M3A7B4089670).

## Author contributions
J.-M.P., T.-W.L, and J.-Y.S. conceived the idea, and developed the materials and methods for the PNC, AINO, and PNC@AINO. J.-M.P., S.H.J., J.P., Y.-H.K., and H.Z. conducted the PL measurements. J.P. and Y.-H.K. conducted the time-resolved PL measurements, and interpreted the results. J.-M.P. and H.Z. designed fully deformable pure green light-emitting devices, and characterized the devices. J.-M.P. and Y.L. designed the schematics, and conducted movie recordings for the experiments. J.-M.P., J.P., Y.-H.K., H.Z., J.M., and J.-Y.S. analyzed the results. J.-M.P. and J.-Y.S. wrote the manuscript with input from all authors. J.-Y.S. supervised the study.

## Competing interests
The authors declare no competing interests.
