## [Peer Review File · Nature Communications]

REVIEWER COMMENTS

Reviewer #1 (Remarks to the Author):

This manuscript reported an aromatic interaction-induced nonpolar organogels (AINOs) as matrix for perovskite nanocrystals. The authors did very detailed investigations on the mechanical properties and stability of AINOs with polystyrene and phenyl octane as components. The resulted mixture of perovskite nanocrystals and AINOs (PNC@AINOs) show very good luminescence properties and stabilities toward various environmental conditions. The result is very interesting. However, there are some issues need to be addressed before consideration for publication in Nature Communications.

Major issues:

1. In this manuscript, the PNC@AINOs work as a light down-conversion layer rather than an emissive layer in light emitting devices. This reduces the impact of this work. Have the authors tried to make LEDs with PNC@AINO as emissive layers?
2. Why the PL peak of V-PNC@AINOs red shifts after polymerization?
3. It is well known that water molecules cause degradation of perovskite. Why the infiltration of water into V-PNC@AINOs cause PL enhancement?
4. The authors claimed that incorporation of PNCs increased the mechanical properties of the AINOs, why?

Minor issues:

1. There are much supporting data in this manuscript. I suggest the authors move all Extended Data Figures to Supplementary information.
2. Figure 1 has limited information; I would suggest the authors delete it.
3. In page 7, line 172, change "Fig. 2b" to "Fig. 3b"

Reviewer #2 (Remarks to the Author):

The author reported a type of stretchable electroluminescent devices consisting of perovskite nanocrystals and nonpolar organogels. For the synthesis of organogel, the effect of solvent on the water solubility, mechanical properties and polarity of organogels was studied through theoretical analysis and experimental research. The obtained perovskite-organogel composite showed excellent PL performance, and had much better tolerance to water than perovskite without an organogel matrix. The author indeed made good progress towards the practical application of perovskite, yet it would be better if the highlight could be clearly presented in this paper.

Major Comments:

Though author did study the nonpolar organogel in detail, the polystyrene organogels have been reported in the previous literatures by using toluene as solvent (e.g., Hiroharu Ajiro et al., *Polymer Journal* (2018) 50:1021–1028). There is no novelty in incorporating perovskite into the organogel or in the device fabrication. The presented originality and novelty of this work unfortunately are not commensurate with the level of Nature Communications. The authors are recommended to mention and discuss it in the introduction part and to highlight the unique advantage of AINO gel developed in this work.

The advantage and application of electric-field driven luminescent devices is unclear as well, considering a very high voltage was applied here.

It is impressive that the organogel matrix with perovskite can be stretched to about 11 times its original length. However, there is limited information about the mechanical property related to the display performance of the flexible light-emitting devices. Since this device is the only practical demonstration of the organogel, what is the advantage of using this organogel over other reported materials in the flexible display devices?

What is the advantage of coupling the blue-light emitter with perovskite, rather than directly driving a green-light emitter?

What is the advantage of using perovskite-organogel composite instead of directly implanting perovskite within the Ecoflex which has been used here by the author to seal the whole device, considering the only function of perovskite is photo-luminescence emitter.

In this article, a green light-emitting bromide based perovskite embedded into the organogel shows high photoluminescence. Recently Iodide based perovskite attracts more attention in related research, due to its poor stability in organic solvent compared to bromide based perovskite, and the red light, together with the green light, can fulfill the basic requirements in displays. Does Iodide based perovskite also show the same excellent performance in this organogel matrix?

What about the durability of the devices, considering the devices operated for only 1 hour in this work, and the long-term test of stretching was not conducted?

From Fig. 5, it seems that the perovskite organogel has very low optical transparency. If the final product has this low transparency, what is the meaning in designing the hydrophobic matrix with such high transparency?

Minor Comments:

Why does M-PNC@AINO decay rapidly in DI water (Fig 2b), but still maintain performance for a long time at both acid and alkaline solution (Fig. 2c).

The plots in the first 5 days in Fig. 2b are indistinguishable.

The quantum yield from the layer of zinc sulfide doped copper/barium titanate to the final green light was not studied.

The energy conversion efficiency of the whole device was not studied.

Reviewer #3 (Remarks to the Author):

In my opinion, this is a very important paper from the scientific and application point of view. The main value of this work is a presentation of a new idea which consists of a synergistic strategy of incorporating perovskite nanocrystals chemically into the highly nonpolar organogels. This idea allowed the authors obtained highly luminescent nanocomposites of the PNCs and the AINOs which characterized by better good luminescence properties and stabilities toward various environmental conditions than previously known.

The work is very extensive and contains a lot of valuable data. Despite this, the publication is read very well because it is written carefully and logically. The results are presented in a clear form and sufficient detail. The title of the paper is correct and reflects the contents of the manuscript and the subheadings accurately determine the results in the paragraphs. The structure and organization of the manuscript are proper. The literature cited is carefully selected and justified. The obtained results were presented in the form of Figures, Extended Data Figures, Extended Data Table, and Supplementary Figures, and Supplementary Video. The films perfectly visualized the properties of the obtained nanocomposites. Each drawing contains a lot of information but is very comprehensively described. The conclusions are adequately supported by the presented data.

I recommend the manuscript for publication without revision.

Responses to Reviewers

We sincerely appreciate the reviewer's valuable comments on our manuscript. The following are point-by-point responses to the reviewer's comments and the changes in the revised manuscript. Sentences inserted into the manuscript are marked in Red.

Reviewer #1

Comment: This manuscript reported an aromatic interaction-induced nonpolar organogels (AINOs) as matrix for perovskite nanocrystals. The authors did very detailed investigations on the mechanical properties and stability of AINOs with polystyrene and phenyl octane as components. The resulted mixture of perovskite nanocrystals and AINOs (PNC@AINOs) show very good luminescence properties and stabilities toward various environmental conditions. The result is very interesting. However, there are some issues need to be addressed before consideration for publication in Nature Communications.

Response: Thank you very much for the reviewer's efforts to review the manuscript. We have carefully considered your comments. Herein, we explain the reviewer's questions and how we revised the paper based on those comments and recommendations.

Comment: In this manuscript, the PNC@AINOs work as a light down-conversion layer rather than an emissive layer in light emitting devices. This reduces the impact of this work. Have the authors tried to make LEDs with PNC@AINO as emissive layers?

Response: Thank you for the reviewer's valuable comment. With increased attention toward advanced technology, recent works have harnessed the development of stretchable devices. For the sake of making a stretchable LED, all layers that constitute the LED have to be stretchable. In other

words, the intrinsically stretchable LED requires intrinsically stretchable form of all layers (including anode, cathode, electron/hole transport layer, etc.) as well as the emissive layer. It is an issue to be solved in stretchable electronics, but remains as a big challenge. Therefore, we tried to demonstrate light emitting devices based on ACEL mechanism with soft ionic conductors, which has lower limitation in mechanical issues. Furthermore, according to the results in the manuscript, chemically anchored PNC in the nonpolar organogels showed high performance as green emitters, whose PLQE reaches near unity after PL enhancement by water post-treatment. It led to highly efficient conversion layers, converting blue light to green light with no luminance decrease under diverse deformations (Fig. 5d-h). Nevertheless, since we also bear in mind that it is important to develop stretchable 'active' perovskite emitter, we will try to make progress toward the issue.

Comment: Why the PL peak of V-PNC@AINOs red shifts after polymerization?

Response: We appreciate the reviewer's constructive question. In fact, we did not define the origin of red shift during polymerization. Nonetheless, similar red shift behaviors of the PNCs have been reported in polymerization process (*ref.* 11, from 508 nm to 522 nm after polymerization) or physically mixing process (*ref.* 15, from 515 nm in solution state to 517 or 518 nm in film state), briefly mentioning that the behavior is due to the residual aggregation of PNCs. So we hypothesized that, when polymerization proceeds, the monomers and the cross-linkers became chemically bonded, which forms solid polymer networks in the PNC solution. These formed networks may cause the slight aggregation of the PNCs, which leads to the increase of the PNC's size, and consequently the emission peak is shifted to red.

Comment: It is well known that water molecules cause degradation of perovskite. Why the infiltration of water into V-PNC@AINOs cause PL enhancement?

Response: Thank you for the reviewer's insightful comment. We pondered and conducted several characterizations when we found the water effect on PL enhancement of the perovskite

nanocomposite. According to the water soaking experiment of the perovskite nanocomposites, the physically dispersed nanocomposite (M-PNC@AINO) underwent initial PL increase followed by gradual decrease. The chemically anchored nanocomposite (V-PNC@AINO) also underwent initial PL increase in first 5 days, however, its PL intensity decreases very slowly, maintaining its initial PL intensities until 110 days. So we hypothesized that, the initial PL enhancement is due to the hydration of metallic Pb, which acts as quenching site, by the infiltrated water molecules into the nonpolar organogels. Then, those water molecules simultaneously decompose the PNC by dissolving it into the precursor state. Therefore, both of the nanocomposites underwent PL enhancement at initial state, but since the ligands in V-PNC@AINO are chemically anchored to the hydrophobic polystyrene chains, meaning that those ligands are hard to be dissolved out into water, the PNC in the V-PNC@AINO can retain its crystal structures in water for long term. This conclusion is supported by the data presented in Supplementary Fig. 22-25.

Comment: The authors claimed that incorporation of PNCs increased the mechanical properties of the AINOs, why?

Response: Thank you for the reviewer's valuable question. The V-PNC is chemically anchored to polymer chains when *in-situ* radically polymerized. From the gel's point of view, this anchored PNC functions as cross-linkers. In addition, a styrenic group in the ligand enables homogeneous dispersion in the gels during polymerization. Therefore, when the synthesized V-PNC@AINO is stretched and undergoes stiffening, the cross-linked points by the PNCs makes the nanocomposite more tough. To make more clear explanation of the V-PNC's role in the AINO from the mechanical point of view, we added a sentence to the manuscript stated as below.

Modification/Addition

1. Modification of the manuscript with additional explanation of the role of V-PNC in the AINO as a cross-linking point.
 - 1) In page 9, line 204.

→ The existence of the PNCs in the nanocomposite does not degrade the mechanical properties of the AINOs, but rather makes the nanocomposite more tough (Fig. 3e and Supplementary Video 2). This toughening behavior is attributed to the role of V-PNC in the nanocomposite; the V-PNC is homogeneously dispersed in the AINO and chemically anchored to polymer chains. That is, the V-PNC behaves as cross-linking point, and therefore it enables the nanocomposite more tough against mechanical stress.

Comment: There are much supporting data in this manuscript. I suggest the authors move all Extended Data Figures to Supplementary information.

Response: Thank you for the reviewer's comment. We move all Extended Data Figures and Tables to Supplementary Information. Then, all mentions about Extended Data Figures and Tables in the manuscript are changed to corresponding Supplementary Figures and Tables.

Comment: Figure 1 has limited information; I would suggest the authors delete it.

Response: We appreciate the reviewer's comment. As the reviewer stated, Figure 1 did not contain all of the results conducted in this work, such as PL enhancement effect or device demonstration. Though, we tried to show the overall visual scheme of this study briefly to a wide range of readers who even may not be familiar with this field. Therefore, we greatly hope that the reviewer considers the preservation of **Figure 1** with a modification as stated below.

Modification/Addition

1. Modification of the **Figure 1** with addition of the nanocomposite's PL enhancement behavior by water.

→

Fig. 1 | Mechanically soft, efficiently luminescent, and environmentally stable perovskite nanocomposites. Synergistic combination of two materials: i) Methylammonium lead bromide (MAPbBr₃) PNCs composed of physically dispersible ligand (M-PNC) or chemically crosslinkable ligand (V-PNC) as highly efficient and pure green emitters, and ii) aromatic interaction-induced nonpolar organogels (AINO) consisting of polystyrene network swollen by phenyl alkanes as passivating matrices for the PNCs. The resulting nanocomposites (PNC@AINOs) complement the luminescence and mechanical property of their parents without trade-off, while their nonpolar environment enhances luminescence stability under various environments, such as air (oxygen), water, acid, base, heat, light, and mechanical stress. **Furthermore, the luminescence efficiency of the PNC@AINOs is increased after water exposure, reaching near unity (~99.8 %).**

Comment: In page 7, line 172, change “Fig. 2b” to “Fig. 3b”

Response: Thank you for the reviewer’s comment. We changed the typo and modified the manuscript stated as below.

Modification/Addition

1. Modification of the manuscript on page 7.

1) In page 8, line 177.

→ The synthesized nanocomposites emit vivid green light under ultraviolet (UV) irradiation (Fig. 3b).

Reviewer #2

Comment: The author reported a type of stretchable electroluminescent devices consisting of perovskite nanocrystals and nonpolar organogels. For the synthesis of organogel, the effect of solvent on the water solubility, mechanical properties and polarity of organogels was studied through theoretical analysis and experimental research. The obtained perovskite-organogel composite showed excellent PL performance, and had much better tolerance to water than perovskite without an organogel matrix. The author indeed made good progress towards the practical application of perovskite, yet it would be better if the highlight could be clearly presented in this paper.

Response: We greatly appreciate the reviewer's effort to review this manuscript. We have carefully considered your comments. Herein, we explain how we revised the paper based on those comments and recommendations.

Comment: Though author did study the nonpolar organogel in detail, the polystyrene organogels have been reported in the previous literatures by using toluene as solvent (e.g., Hiroharu Ajiro et al., *Polymer Journal* (2018) 50:1021–1028). There is no novelty in incorporating perovskite into the organogel or in the device fabrication. The presented originality and novelty of this work unfortunately are not commensurate with the level of Nature Communications. The authors are recommended to mention and discuss it in the introduction part and to highlight the unique advantage of AINO gel developed in this work.

Response: We appreciate the reviewer's considerate comment on the value of this manuscript in the context of existing literature. As the reviewer mentioned, polystyrene organogels are not firstly reported in this paper. Several studies have reported swelling characteristics of the organogels in good solvents, including poly(styrene-co-divinylbenzene) in toluene (R. F. Boyer & R. S. Spencer, (1948) *J. Polym. Sci.* 3, 97-127), or in limonene with polyelectrolyte additives (H. Ajiro et al., *Polymer Journal*

(2018) 50:1021–1028), or poly(methyl methacrylate) (PMMA) in toluene (F. Doumenc *et al.*, *The European Physical Journal E* (2008) 27, 3-11). Although those papers studied the organogel system based on nonpolar materials, they did not focus on the ‘nonpolar’ property of the organogels itself, needless to say that they have not considered a stretchable form for soft device applications. The ionic-bonded perovskite nanocrystals have low formation energy, which means that they are highly vulnerable to polar environment. Therefore, the nonpolar environment is necessitated for the stable formation and preservation of perovskite emitters, and from the point, we conceptualize the stretchable ‘nonpolar’ organogels. We think that the thought of combination of the perovskite nanocrystals with the nonpolar organogels is the important insight in materials science. The reviewer claimed that there is less novelty in this work, because the organogels based on nonpolar materials exist. However, the conceptualization of nonpolar organogels with desired properties, and their combination with the perovskite emitters itself is the significance of this paper, which has not been accomplished to the best of our knowledge. Therefore, we cannot agree with the reviewer’s claim on the novelty of this paper.

Furthermore, previously reported perovskite-elastomer nanocomposites contain the PNC as a physically mixed and dispersed form, which has a limitation on closer affinity between the PNC and the elastomers, therefore their PLQE is relatively low (for example, ~75 % for *ref.* 15, or ~87 % for *ref.* 12, but not self-standing form). This work proposed the methodology of *in-situ* chemically anchoring the PNC to the nonpolar organogels, which led to high softness and stretchability, stability, and more importantly, the PL enhancement behavior of the nanocomposites after water exposure that materializes remarkably efficient elastomeric perovskite emitters, having record PLQE of near unity (~99.8 %) for the first time.

We believe these are the significance of the manuscript, but with the reviewer’s comment we concluded the introduction part did not explain or highlight the points enough. Therefore, we modified the manuscript with additional texts in introduction part as stated below.

Modification/Addition

1. Modification of the introduction for elaborating the significance of the manuscript.

1) In page 2, line 49.

→ While elastomeric nanocomposites have also been reported^{12,13,14,15}, ~~the PNC is physically mixed and dispersed in those matrices, therefore~~ their luminescence efficiency ~~and~~ or environmental stability ~~has~~ remained insufficient for practical application.

2) In page 3, line 63.

→ ~~Some studies have reported swelling behaviors of the organogels, including the system of polystyrene (PS) in toluene¹⁹, or in limonene with assistance of polyelectrolyte²⁰, or poly(methyl methacrylate) (PMMA) in toluene or other good solvents²¹. However, the ‘nonpolar’ property of the organogels, which is an important requisite for the stable formation of PNCs, has little been considered. Therefore, a conception that combines perovskite ionic crystals with nonpolar organogels, which could materialize a mechanically soft perovskite emitter, has also not been studied so far.~~

3) In page 3, line 71.

~~→ Introducing aromatic non-covalent interaction into the nonpolar organogels is accomplished with general theoretical and experimental consideration of the polymer-solvent system.~~ ~~The mechanical, optical, and nonpolar properties of the organogels are carefully optimized with the theoretical and experimental considerations of polymer-solvent system.~~ Taking advantage of the AINO's nonpolar characteristic, luminescent green nanocomposites are synthesized by physically or chemically combining the MAPbBr₃ PNCs with the AINOs.

2. Addition of the references related to the previous studies on the nonpolar organogels.

18. Shi, L. *et al.* Dielectric gels with ultra-high dielectric constant, low elastic modulus, and excellent transparency. *NPG Asia Mater.* **10**, 821-826 (2018).

19. Boyer, R. F., & R. S. Spencer. Some thermodynamic properties of slightly cross-linked styrene-divinylbenzene gels. *J. Polym. Sci.* **3**, 97-127 (1948).

20. C., Preeyarad, & H. Ajiro. The electrostatic advantages of cross-linked polystyrene organogels swollen with limonene for selective adsorption and storage of hydrophobic drugs. *Polym. J.*, **50**, 1021-1028 (2018).
21. F. Doumenc, H. Bodiguel, and B. Guerrier. Physical aging of glassy PMMA/toluene films: Influence of drying/swelling history. *Eur. Phys. J. E* **27**, 3-11 (2008).
22. Flory, P. J. *Principles of polymer chemistry*. (Cornell University Press, 1953).

Comment: The advantage and application of electric-field driven luminescent devices is unclear as well, considering a very high voltage was applied here.

Response: Thank you for the reviewer's comment. As the reviewer mentioned, the ACEL device based on ZnS:Cu phosphors needs high voltage for the operation. However, our purpose on the device demonstration is the applicability of V-PNC@AINO as a 'color conversion layer', meaning that any light source can be a backlight, such as LED, OLED, etc. For the fully stretchable form, we just adopted stretchable ACEL layer as a blue light source, which is well established in previous studies. Therefore, we hope that the reviewer focuses on the properties of V-PNC@AINO as a color conversion layer. We did not claim our novelty of this paper on the ACEL device structure.

Comment: It is impressive that the organogel matrix with perovskite can be stretched to about 11 times its original length. However, there is limited information about the mechanical property related to the display performance of the flexible light-emitting devices.

Response: We appreciate the reviewer's valuable comment. As we mentioned above, we did not claim the originality on the ACEL device based on doped ZnS phosphors. We claimed the superior properties of the V-PNC@AINO with water post-treatment as a color conversion layer. Besides, the consideration of the device's mechanical property related with display performance is actually established by the ACEL device mechanism, not by the V-PNC@AINO. We conducted mechanical

tensile tests of the devices during operation. As shown in Supplementary Fig. 36, the device can be stretched to 2.7 times without interfacial failure. When the device is stretched over 2.7, the layers between ACEL nanocomposite ($\text{ZnS}:\text{Cu}-\text{BaTiO}_3@\text{Ecoflex}$) and encapsulating matrix (Ecoflex 00-30) is delaminated. Since we fabricated the ACEL layer with high content of $\text{ZnS}:\text{Cu}$ phosphor and BaTiO_3 powders for efficient electroluminescence, the ACEL layer slightly sacrifices its stretchability, consequently leading to the delamination at relatively low stretch.

All constituents of the device are soft, where the elastic modulus of the hydrogel conductor, blue ACEL layer, color conversion layer, encapsulating matrix, and the integrated device is 5.4 kPa, 152.1 kPa, 111.6 kPa, 52.6 kPa and 50.4 kPa, respectively.

The luminance of the device is increased during stretch, because the thickness of ACEL layer is decreased but the applied voltage is constant, meaning that the applied electric field on the ACEL layer is increased. The color coordinate of the device is slightly changed during stretch, but still maintains its high purity as stated below.

Modification/Addition

1. Addition of Supplementary Fig. 35 for the mechanical properties of the device related to its display performance

→

Supplementary Fig. 35. Display performance of the device with mechanical properties. a, Tensile stress-stretch curves of the integrated device and its constituents, including PAAm hydrogel conductor, V-PNC@AINO, ACEL nanocomposite ($\text{ZnS}:\text{Cu}-\text{BaTiO}_3@\text{Ecoflex}$), and encapsulating matrix Ecoflex

00-30. The interface between ACEL nanocomposite and encapsulating matrix is delaminated when the stretch reaches about 2.7. **b**, Luminance of the device as a function of applied stretch. Values represent the mean and standard deviation ($n=3$). **c**, Color coordinates of the emission light of the device under stretch. The applied voltage and frequency is 2.12 kV_{rms} and 4 kHz, respectively.

2. Modification of the manuscript for the mechanical properties of the device related to its display performance.

1) In page 130, line 300.

→ Since all the constituents of the device are soft, it has the elastic modulus of ~52 kPa (Supplementary Fig. 35a). Therefore, we demonstrate a manual operation of the device, emitting bright and vivid green light under diverse deformations such as bending, folding, stretching, and twisting without device failure, delamination, or luminescent hysteresis (Fig. 5e–h and Supplementary Video 5).

2) In page 13, line 309.

→ The color ~~purity~~ coordinate of green light emitted from the devices is slightly ~~decreases~~ changed under deformation (Supplementary Fig. 31c, 35c), but still remains pure, exceeding the NTSC color standard.

Comment: Since this device is the only practical demonstration of the organogel, what is the advantage of using this organogel over other reported materials in the flexible display devices?

Response: We appreciate the reviewer's comment. The ideal stretchable perovskite color conversion layer must combine high photoluminescence quantum efficiency (PLQE) and optical density (absorbance) for efficient color conversion, high stretchability with mechanical softness for low-constrained deformability, and high environmental stabilities for practical application. The high PLQE and stability can be achieved when the encapsulating matrix has high nonpolarity, which enables more affinitive interaction with the PNC. The high optical density is achieved when the

encapsulating matrix has high solubility of the PNC. The high softness and stretchability of polymeric materials can be realized with careful component selection and compositional engineering. Therefore, the encapsulating matrix for the PNC must be simultaneously nonpolar and soft. Existing reported materials (especially polymers) for the PNC encapsulating matrices didn't possess the above properties at the same time. The highly nonpolar matrices (*ref.* 9-11) were often rigid or brittle, even though their PLQE is relatively high (~90 %). The stretchable matrices were suffered from low PLQE due to insufficient nonpolarity (*Nanophotonics*, (2018), 7, 1949-1958 or *ref.* 14 in the manuscript), not self-standing and weak mechanical properties (*ref.* 12), or insufficient combination of stretchability, softness, and PLQE (*ref.* 13 : stretchability ~170 %, modulus ~1 MPa, PLQE ~60 %, *ref.* 15 : stretchability ~500 %, modulus ~1 MPa, PLQE ~70 %). In this work, the aromatic nonpolar organogels containing highly concentrated PNCs were remarkably soft, stretchable, and dense (elastic modulus ~100 kPa, stretchability ~1000 %, and absorbance ~2.5 at 450 nm). Furthermore, when the PNCs were chemically anchored by in-situ polymerized with the organogel precursors, the nanocomposite showed high PLQE (~92 %) and reached near unity after water exposure (~99.8 %), where these combined superior properties for color conversion layer has not been reported as far as we know. Thus, the V-PNC@AINO was essential for the efficient stretchable color conversion layer in the device. We added the consideration of the necessity for introducing V-PNC@AINO as a color conversion layer in Supplementary Notes as stated below.

Modification/Addition

1. Addition of the paragraph of the necessity for introducing V-PNC@AINO as a color conversion layer in Supplementary Notes.

→ **4. The V-PNC@AINO as an efficient color conversion layer.** The ideal stretchable perovskite color conversion layer must combine high PLQE and optical density (absorbance) for efficient color conversion, and high stretchability with mechanical softness for low-constrained deformability. We compare the V-PNC@AINO with a nanocomposite consisting of the V-PNC and poly(dimethyl siloxane) (PDMS), which is commercially available siloxane based transparent

elastomer, similar to Ecoflex 00-30. For the nanocomposite based on the V-PNC and PDMS (V-PNC@PDMS), when the concentration of PNC is over 4 mg/ml, the precursor mixture becomes yellowish, which means an aggregation occurred. Even for lower concentration (~1 mg/ml), the PLQE of the V-PNC@PDMS is relatively low (~40 %) compared to the V-PNC@AINO with high concentration (~12 mg/ml) (Supplementary Fig. 37). Also, the V-PNC@PDMSs are environmentally unstable, which lose their luminescence properties ~6 days in air, water, and ~12 hours under UV light. Most importantly, the color coordinate of the device based on V-PNC@PDMS as a color conversion layer is (0.197, 0.390), which has relatively low color purity than that based on the V-PNC@AINO, due to its low optical density and PLQE (Supplementary Fig. 28).

Meanwhile, the V-PNC@AINO efficiently converts blue light to green in the device. As shown in Fig. 4b and Supplementary Fig. 27, the water soaking affects the optical behavior of V-PNC@AINO, where both of the absorbance and PLQE are increased. Therefore, when we use the V-PNC@AINO after 7 days in water as a color conversion layer, the color coordinate of the device is changed from (0.210, 0.625) to (0.192, 0.752), which coordinate is comparable to that of QD-LEDs and Pe-LEDs.

2. Addition of Supplementary Fig. 28 for the necessity of introducing V-PNC@AINO as a color conversion layer.

→

Supplementary Fig. 28. Water effect on the V-PNC@AINO as an efficient color conversion layer.

a, Luminance of the devices with different color conversion layer, including pristine blue ACEL layer, V-PNC with PDMS (V-PNC@PDMS), pristine V-PNC@AINO, and V-PNC@AINO soaked in water for 7 days. The applied electric field and frequency is 3 V/ μ m and 4 kHz, respectively. **b**, Color coordinates of the devices with different color conversion layer, including pristine blue ACEL layer, V-PNC with PDMS (V-PNC@PDMS), pristine V-PNC@AINO, and V-PNC@AINO soaked in water for 7 days.

Comment: What is the advantage of coupling the blue-light emitter with perovskite, rather than directly driving a green-light emitter?

Response: Thank you for the reviewer's insightful comment. To drive electroluminescence directly from the perovskite emitters, a device must possess the structure of conventional LED. For the fabrication of a stretchable LED, all layers that constitute the LED have to be stretchable. In other words, the intrinsically stretchable LED requires intrinsically stretchable form of all layers (including anode, cathode, electron/hole transport layer, etc.) as well as the emissive layer. It is an issue to be solved in stretchable electronics, but remains as a big challenge. Therefore, we tried to demonstrate light emitting devices based on ACEL mechanism with soft ionic conductors which has low limitation

in mechanical issues. Furthermore, according to the results in the manuscript, chemically anchored PNC in the nonpolar organogels showed high performance as green emitters, whose PLQE reaches near unity after PL enhancement by water post-treatment. It led to highly efficient conversion layers, converting blue light to green light with no luminance decrease under diverse deformations (Fig. 5d-h). Nevertheless, since we also bear in mind that it is important to develop stretchable 'active' perovskite emitter, we will try to make progress toward the issue.

Comment: What is the advantage of using perovskite-organogel composite instead of directly implanting perovskite within the Ecoflex which has been used here by the author to seal the whole device, considering the only function of perovskite is photo-luminescence emitter.

Response: We appreciate the reviewer's reasonable comment. As we mentioned above the reviewer's comment, the most important requisites for superior perovskite color conversion layer are its high PLQE and optical density. We previously fabricated perovskite nanocomposites based on the V-PNC and poly(dimethylsiloxane) (PDMS), which is transparent elastomer, and a type of siloxane based polymer similar to Ecoflex 00-30. Then we measured their PLQE and relative PL intensities in ambient air, DI water, and UV light under same experimental condition with the PNC@AINO as shown in Supplementary Fig. 37. Firstly, when the PNC concentration was over 4 mg/ml, the PNC and PDMS precursor mixture became yellowish, which means severe aggregation occurred. Even for lower concentration (~1 mg/ml) the absolute PLQE of the V-PNC@PDMS is relatively low (~40%) compared to the V-PNC@AINO with high concentration (~12 mg/ml). Furthermore, the V-PNC@PDMSs were environmental unstable, which lost their luminescence properties ~6 days in air, water, and ~12 hours under UV light. Therefore, the use of V-PNC@AINO was essential for the efficient PL conversion emitter in the device. As a comparison for the performance of perovskite nanocomposite, we added Supplementary Fig. 37 for the stability and PL properties of the V-PNC@PDMS and explanation of fabrication method in Methods and Supplementary Information.

Modification/Addition

1. Addition of fabrication method of the PNC@PDMS nanocomposite in Methods section.

1) In page 19, line 442.

→ To compare the performance as perovskite nanocomposite emitters, we fabricated the nanocomposite based on the V-PNC and the poly(dimethylsiloxane) (PDMS, Sylgard 184, Dow Corning) which is transparent elastomer, and a type of siloxane based polymer similar to Ecoflex 00-30. We mixed 10 ml of the V-PNC solution (~1 mg/ml in toluene) with 10 g of PDMS mixture (10:1 weight ratio of precursor and curing agent), dried toluene under vacuum, and poured the mixture onto glass mold with 1 mm thickness. After cured overnight at 60 °C in the glove box, the V-PNC and PDMS nanocomposite (V-PNC@PDMS) sample was obtained.

2. Addition of Supplementary Fig. 37 for the PL properties and environmental stability test of PNC-PDMS composite.

→

Supplementary Fig. 37. Environmental stability of the V-PNC@PDMS nanocomposites. a, Relative PL intensity and PLQE of the V-PNC@PDMS in ambient air. **b,** Relative PL intensity and PLQE of the V-PNC@PDMS in DI water. **c,** Relative PL intensity and PLQE of the V-PNC@PDMS under UV irradiation.

3. Addition of the consideration for introducing V-PNC@AINO as a color conversion layer in

Supplementary Notes.

→ **4. The V-PNC@AINO as an efficient color conversion layer.** The ideal stretchable perovskite color conversion layer must combine high PLQE and optical density (absorbance) for efficient color conversion, and high stretchability with mechanical softness for low-constrained deformability. We compare the V-PNC@AINO with a nanocomposite consisting of the V-PNC and poly(dimethyl siloxane) (PDMS), which is commercially available siloxane based transparent elastomer, similar to Ecoflex 00-30. For the nanocomposite based on the V-PNC and PDMS (V-PNC@PDMS), when the concentration of PNC is over 4 mg/ml, the precursor mixture becomes yellowish, which means an aggregation occurred. Even for lower concentration (~1 mg/ml), the PLQE of the V-PNC@PDMS is relatively low (~40 %) compared to the V-PNC@AINO with high concentration (~12 mg/ml) (Supplementary Fig. 37). Also, the V-PNC@PDMSs are environmentally unstable, which lose their luminescence properties ~6 days in air, water, and ~12 hours under UV light. Most importantly, the color coordinate of the device based on V-PNC@PDMS as a color conversion layer is (0.197, 0.390), which has relatively low color purity than that based on the V-PNC@AINO, due to its low optical density and PLQE (Supplementary Fig. 28).

Meanwhile, the V-PNC@AINO efficiently converts blue light to green in the device. As shown in Fig. 4b and Supplementary Fig. 27, the water soaking affects the optical behavior of V-PNC@AINO, where both of the absorbance and PLQE are increased. Therefore, when we use the V-PNC@AINO after 7 days in water as a color conversion layer, the color coordinate of the device is changed from (0.210, 0.625) to (0.192, 0.752), which coordinate is comparable to that of QD-LEDs and Pe-LEDs.

Comment: In this article, a green light-emitting bromide based perovskite embedded into the organogel shows high photoluminescence. Recently Iodide based perovskite attracts more attention in related research, due to its poor stability in organic solvent compared to bromide based perovskite,

and the red light, together with the green light, can fulfill the basic requirements in displays. Does Iodide based perovskite also show the same excellent performance in this organogel matrix?

Response: Thank you for the reviewer's meaningful comment. As the reviewer mentioned, it is important to materialize stretchable and stable perovskite pure red emitters. Unfortunately, the iodide based perovskite nanocrystals are not stable themselves in solution state, due to crystal phase instability from larger anion size (Zhang et al., *ACS Nano* (2015) **9**, 4532-4542). To overcome the issue, considerable studies have been investigated to enhance the intrinsic PL properties, including cation compositional engineering, ligand engineering, salt or ligand post-treatment, etc. If the red perovskite emitter itself were fabricated to be stably dispersible in nonpolar solvents with high PLQE, it could be compatible with the AINO, because the fabrication process of synthesizing PNCs is independently preceded before *in-situ* AINO polymerization. Even though, we greatly hope that the reviewer considers the applicability of the AINOs confined to green perovskite emitters, in order not to have the manuscript too extensive. As we agreed the reviewer's opinion, the demonstration of stable and efficient red perovskite emitters with stretchable form should be studied in future works to achieve a wide color gamut of the deformable light-emitting devices.

Comment: What about the durability of the devices, considering the devices operated for only 1 hour in this work, and the long-term test of stretching was not conducted?

Response: We appreciate the reviewer's reasonable comment. We conducted the device durability test by measuring the luminance of the device during 1000 times stretch-release. Since the operating voltage is high, we can't operate the voltage amplifier for a long time due to the safety issue. Instead, we can operate the device with time interval, measuring the luminance change under exposure of the device in ambient air. As shown in Supplementary Fig. 35, the luminance of the device is almost maintained during stretch-release cycle test and after 20 days in the air. We added the durability information of the device in Supplementary Information and modified the main text as stated below

Modification/Addition

1. Addition of Supplementary Fig. 35 for the consideration of the device durability.

→

Supplementary Fig. 36. Durability of the device. a, Luminance of the device after cyclic stretch-release test. The cyclic stretch is 2. **b,** Luminance of the device with time interval in ambient air. The applied voltage and frequency for both of the durability tests are fixed to 3 V/μm and 4 kHz, respectively.

2. Modification of the main text with the mention of the device durability.

- 1) In page 13, line 304.

→ Furthermore, the device maintains its luminance for 1 hour in operation, cyclic stretch-and-release operation, or operation with time interval (Supplementary Fig. 31c, 36).

Comment: From Fig. 5, it seems that the perovskite organogel has very low optical transparency. If the final product has this low transparency, what is the meaning in designing the hydrophobic matrix with such high transparency?

Response: We appreciate the reviewer's meaningful comment. We tried to approach the

development of soft elastomer with the concept of nonpolar organogels, which led to the fabrication of materials that possess stretchability, transparency, and hydrophobicity with additional softness, comparable to existing elastomers. Therefore, the aromatic nonpolar organogels have potentials to be utilized as a soft dielectric elastomer or scaffolds for any nonpolar materials, as well as perovskite nanocrystals. The reason we considered the transparency of nonpolar organogels is enhancing the potential applicability as a new soft platform. Furthermore, if the nonpolar organogels became opaque or translucent, it would decrease the luminescence efficiency of the nanocomposite by absorbing light emitted from the perovskite nanocrystals.

Comment: Why does M-PNC@AINO decay rapidly in DI water (Fig 2b), but still maintain performance for a long time at both acid and alkaline solution (Fig. 2c).

Response: Thank you for the reviewer's comment. The data plotted in Fig. 4c is respectively the PL intensity and the PLQE of the V-PNC@AINO in the acidic and basic environment, not those of the M-PNC@AINO. Because the M-PNC@AINO had lower stability toward DI water, we did only conduct the acid (or base) stability of the V-PNC@AINO. The water stability and highly nonpolar environment of the V-PNC@AINO enables the acquisition of acidic and basic resistance.

Comment: The plots in the first 5 days in Fig. 2b are indistinguishable.

Response: We appreciate the reviewer's careful comment. We changed the time period shown in Fig.4b and 4c to log scale for visual clearance as stated below.

Modification/Addition

1. Modification of the time period of Fig. 4b and 4c to log scale for visual clearance.

→

Comment: The quantum yield from the layer of zinc sulfide doped copper/barium titanate to the final green light was not studied.

Response: We appreciate the reviewer’s meaningful comment. As we measured the luminance of the devices, we calculated the color conversion efficiency of the devices by dividing the luminance with perovskite emitting layer into the luminance without perovskite emitting layer, as shown in Supplementary Fig. 32. We found that the color conversion efficiency of the device is about 97 % in average.

Modification/Addition

1. Addition of Supplementary Fig. 32 for the consideration of the color conversion efficiency.

→

Supplementary Fig. 32. Color conversion efficiency of the devices as a function of applied electric field. The color conversion efficiency is calculated by dividing the luminance of the device with the V-PNC@AINO layer into the luminance of the device without the V-PNC@AINO layer.

2. Addition of the paragraph in Supplementary Notes for the consideration of the color conversion efficiency.

→ **5. Color conversion efficiency of the device.** We calculate the color conversion efficiency of the device by dividing the luminance with color conversion layer into the luminance without color conversion layer. As shown in Supplementary Fig. 32, the efficiency of the devices is about 97 % regardless of the applied electric field, due to the superior optical properties of the water-induced PL enhanced V-PNC@AINO.

3. Modification of the main text for the mention of the color conversion efficiency of the device.

1) In page 12, line 293.

→ With increasing electric field, the luminance also increases for both of the devices with and without the color conversion layer. **The color conversion efficiency of the device is about 97 % regardless of the applied electric field (Supplementary Notes 5 and Supplementary Fig. 32).**

Comment: The energy conversion efficiency of the whole device was not studied.

Response: We appreciate the reviewer's comment. As we mentioned above, the focus on this study is the applicability of V-PNC@AINO as a color conversion layer, not the ACEL device operation. We thought that the consideration of ACEL device's energy conversion efficiency has already been well established by previous paper (*ref. 4*), and is beyond the scope of our study, nonetheless, we conducted a series of measurement and calculations. Since the device emits light from electric signal, we measured the luminous power and the consumed power of the whole device system. First, the luminous power of the device at $3\text{ V}/\mu\text{m}$ is about $8.69 \times 10^{-2}\text{ lm}$, which value is calculated by multiplying the measured luminance ($34.3\text{ cd}/\text{m}^2$) with the measured emission area ($7.85 \times 10^{-5}\text{ m}^2$), and dividing it by the steradian (0.031 sr), calculated from the measurement setup. Then, we obtained the consuming power of the device by measuring the current and the phase difference with the applied voltage. We observed the current through the device with the oscilloscope (DPO5104B, Tektronix) and the low-noise current preamplifier (Stanford research systems, SR570). As shown in Supplementary Fig. 34b, when the applied root-mean-square voltage V_{RMS} is 2.12 kV , the root-mean-square current I_{RMS} is 0.53 mA . The phase difference is 79.2° , which value is calculated by dividing the time difference at the peak position of the voltage and the current ($5.5 \times 10^{-5}\text{ s}$) into the period ($2.5 \times 10^{-4}\text{ s}$), and then multiplying 360° . Since the power $P = V_{\text{RMS}} I_{\text{RMS}} \cos \phi$, therefore, the consumption power of the device is 0.21 W . Finally, the luminous efficacy is obtained by dividing the luminous power into the consumption power, which value is about $413.6\text{ mlm}/\text{W}$. We added the above consideration for the power conversion efficiency of the device in the Main text and Supplementary Information as stated below.

Modification/Addition

1. Addition of Supplementary Fig. 34 for the power conversion efficiency of the device in Supplementary Information.

→

a

b

Supplementary Fig. 34. Power consumption of the device. **a**, Electric circuit for the measurement of power consumption of the device. The voltage is applied to the device by the function generator with the high voltage amplifier, while the current and the phase shift is measured by the oscilloscope with the current preamplifier. **b**, Applied voltage and the measured current as a function of time. The applied frequency and the voltage is 4 kHz and 6 kV_{pp}, respectively. The time difference at peak position of the voltage and the current is $5.5 \times 10^{-5} \text{ s}$, therefore the phase shift is calculated to 79.2° . The power consumption is 0.21 W, obtained by the following equation, $P = V_{RMS} I_{RMS} \cos \phi$, where V_{RMS} and I_{RMS} are the voltage and the current of root-mean-square value, and $\cos \phi$ is the phase shift.

2. Addition of the consideration of the power conversion efficiency of the device in Supplementary Notes.

→ **6. Power conversion efficiency of the device.** Since the device emits light from electric signal, we measured the luminous power and the consumed power of the whole device system. First, the luminous power of the device at 3 V/ μm and 4 kHz is calculated to $8.69 \times 10^{-2} \text{ lm}$, which value is obtained by multiplying the measured luminance (34.3 cd/m^2) with the measured emission area ($7.85 \times 10^{-5} \text{ m}^2$), and dividing it into the steradian (0.031 sr), calculated from the measurement setup.

Then, we calculate the consuming power of the device by measuring the current and the phase difference with the applied voltage. We observe the current by connecting the device with the oscilloscope (DPO5104B, Tektronix) and the low-noise current preamplifier (Stanford research systems, SR570) in series (Supplementary Fig. 34a). As shown in Supplementary Fig. 34b, when the applied root-mean-square voltage V_{RMS} is 2.12 kV, the root-mean-square current I_{RMS} is 0.53 mA. The phase difference is 79.2° , which value is calculated by dividing the time difference at the peak position of the voltage and the current (5.5×10^{-5} s) into the period (2.5×10^{-4} s), and then multiplying 360° . Since the electric power P is equal to $V_{\text{RMS}} I_{\text{RMS}} \cos \phi$, the consumption power of the device is 0.21 W. Finally, the luminous efficacy is obtained to 413.6 mlm/W, by dividing the luminous power into the consumption power. The obtained luminous power value is somewhat higher than previous study⁴, possibly due to the existence of BaTiO₃ and larger amount of ZnS:Cu phosphors in the ACEL layer.

3. Modification of the main text for the mention of the power conversion efficiency of the device.

1) In page 13, line 298.

→ The power conversion efficiency of the device is about 413.6 mlm/W (Supplementary Notes 6 and Supplementary Fig. 34).

4. Addition of the measurement procedure in Methods for obtaining power consumption of the device.

1) In page 24, line 571.

→ To measure the power consumption of the device, we observed the current by connecting the device with an oscilloscope (DPO5104B, Tektronix) and a low-noise current preamplifier (Stanford research systems, SR570) in series, while applying AC voltages of 2.12 kV_{pp} with 4 kHz.

Reviewer #3

Comment: In my opinion, this is a very important paper from the scientific and application point of view. The main value of this work is a presentation of a new idea which consists of a synergistic strategy of incorporating perovskite nanocrystals chemically into the highly nonpolar organogels. This idea allowed the authors obtained highly luminescent nanocomposites of the PNCs and the AINOs which characterized by batter good luminescence properties and stabilities toward various environmental conditions than previously known.

Response: Thank you for the reviewer's affirmative comment for the value of the manuscript. We have made efforts to develop a synergistic concept for soft perovskite nanocomposites with unconventional gels, and we are pleased that reviewer #3 gave a good evaluation on this idea in the manuscript.

Comment: The work is very extensive and contains a lot of valuable data. Despite this, the publication is read very well because it is written carefully and logically. The results are presented in a clear form and sufficient detail. The title of the paper is correct and reflects the contents of the manuscript and the subheadings accurately determine the results in the paragraphs. The structure and organization of the manuscript are proper. The literature cited is carefully selected and justified. The obtained results were presented in the form of Figures, Extended Data Figures, Extended Data Table, and Supplementary Figures, and Supplementary Video. The films perfectly visualized the properties of the obtained nanocomposites. Each drawing contains a lot of information but is very comprehensively described. The conclusions are adequately supported by the presented data.

Response: We sincerely appreciate the reviewer's considerate effort to review the manuscript. We tried to organize the concept of nonpolar organogels as a counterpart scaffolds for soft devices with careful consideration, which led to the manuscript to include a large amount of data. We appreciate

again for the reviewer's comprehensive review for the manuscript.

Comment: I recommend the manuscript for publication without revision.

Response: We appreciate the reviewer's support for publication of the manuscript on *Nature Communications*. Since we had several meaningful comments from other reviewers, we have modified the manuscript with additional data or texts. It would be greatly helpful for us if the reviewer gave any comment for the revised manuscript.

Additional modification

1. Modification of reference numbers in Supplementary Fig. 31c.

→

Supplementary Fig. 31. Luminescence properties of the fully deformable pure green light-emitting devices. **a**, Color coordinates of PL emission of the color conversion layer (V-PNC@AINO), electroluminescence of the blue ACEL layer (ZnS:Cu-BaTiO₃@Ecoflex), and electroluminescence of the device on the CIE 1931 color space. **b**, PL spectra and PLQE values of the V-PNC@AINO soaked in water for 7 days as a color conversion layer, which is excited by the wavelength of blue light series from 365 to 485nm. **c**, Color coordinates of the device in various deformed state and after 1 hour operation, plotted together with color coordinates of the NTSC TV color standard, and reported studies of Pe-LEDs^{33,34,35,36} and QD-LEDs^{37,38} on the CIE 1931 color space.

2. Deletion of a word in page 1, line 25.

→ ultraviolet light,

3. Typo: In page 34, line 751 of Fig. 5.

→ ~~zn~~zinc

4. Modification of the manuscript in page 3, line 75.

→ The **chemically anchored, and** concentrated nanocomposite is brightly luminescent that it shows considerable luminescence efficiency (up to 99.8 %) after water exposure, **while possessing** high softness and stretchability (Young's modulus of ~100 kPa, and stretchability of ~11), which performance in the form of elastomeric perovskite emitters has not previously been reported ~~as far as we know~~ **to the best of our knowledge.**

5. Modification of the manuscript in page 8, line 176.

→ We then synthesize two types of highly luminescent nanocomposites of the PNCs and the AINOs (M- and V-PNC@AINO) by *in-situ* polymerization (**See Methods**), where the solvent

and $\phi_{p,v}$ of the AINO are chosen to phenyl octane and 0.65, respectively (Supplementary Fig. 4b).

6. Modification of the manuscript in page 11, line 266.

→ The outstanding luminescence efficiency, compliant mechanical property, and environmental stability of the V-PNC@AINO enables its practical application to a soft light-emitting layer in **recently developed soft ISELD system**⁴.

7. Modification of the manuscript in page 11, line 269.

→ We use **a typical** stretchable ACEL layer based on PAAm hydrogel conductor, and electroluminescent composites of zinc sulfide doped copper (ZnS:Cu) and barium titanate (BaTiO₃) in Ecoflex 00-30 (ZnS:Cu-BaTiO₃@Ecoflex 00-30) as a blue light source.

8. Modification of the manuscript in page 13, line 320.

→ Its elastic modulus decreases from gigapascals to kilopascals by just 0.3 volume ratio of the solvent, which endows widely tunable mechanical behavior of the **nonpolar** organogels.

REVIEWERS' COMMENTS:

Reviewer #1 (Remarks to the Author):

In the revised manuscript and response letter, the authors have addressed some of my comments. However, I think there are two issues still need to be addressed.

1. I think PNC@AINO as light conversion layer lacks novelty. I insist that demonstration of LEDs with PNC@AINO as light emitting layers is critical for this work. I understand that, as the authors mentioned, it is challenging to make stretchable LEDs. However, the authors should try to make regular LEDs on ITO substrate with PNC@AINO as light emitting layers.
2. As for the PL red shifts in polymerized V-PNC@AINOs, it is expected that the polymer networks suppress aggregation of the PNCs. Why the polymer networks cause aggregation of PNCs? Can the author provide evidence?

Reviewer #2 (Remarks to the Author):

The author has answered our questions very well, and the novelty and the necessity of the preparation strategies has also been highlighted and discussed in detail. It is recommended for publication as is.

Reviewer #3 (Remarks to the Author):

I accept the changes introduced in the publication in accordance with the comments of other reviewers. These changes made it possible to improve the publication that I find very interesting and valuable

Responses to Reviewers

Reviewer #1

Comment: In the revised manuscript and response letter, the authors have addressed some of my comments. However, I think there are two issues still need to be addressed.

Response: Thank you for the reviewer's effort to review the revised manuscript. We have considered your insightful comments. Herein, we answer the reviewer's comments and questions as stated below.

Comment: I think PNC@AINO as light conversion layer lacks novelty. I insist that demonstration of LEDs with PNC@AINO as light emitting layers is critical for this work. I understand that, as the authors mentioned, it is challenging to make stretchable LEDs. However, the authors should try to make regular LEDs on ITO substrate with PNC@AINO as light emitting layers.

Response: We greatly appreciate the reviewer's comment. As the reviewer mentioned, the direct electroluminescence (EL) operation for light-emitting device is important. To operate EL device, however, a light-emitting layer is designed with the conductive pathway for charge injection (electron from anode and hole from cathode) to the light-emitting layer. Our approach, on the other hand, is designed to insulating perovskite nanocrystals from external stimuli. The AINO is highly nonpolar materials, which means that it behaves as a low-k dielectric insulator. By insulating the PNCs with the AINO, instead, we materialize the nanocomposites that possess highly efficient photoluminescence (PL) properties and considerable stabilities toward various conditions. To design materials that have conductive pathway with good PL properties and stabilities requires considerable effort and time, and we think it is beyond the purpose of this paper.

Nevertheless, we appreciate and agree with the reviewer's comment, where the active operation of

PNCs as an electroluminescence source is important. Related materials fabrication and experiments for EL operation will be conducted in follow-up study. Therefore, we sincerely hope that the reviewer considers this point.

Comment: As for the PL red shifts in polymerized V-PNC@AINOs, it is expected that the polymer networks suppress aggregation of the PNCs. Why the polymer networks cause aggregation of PNCs? Can the author provide evidence?

Response: Thank you for the reviewer's comment. Perovskite nanocrystal is one of the types of quantum dot, which means that it has dimensions of nanometer scale, so its optical behavior is explained with quantum confinement effect. The size of PNCs, therefore, affects their emission wavelength because of the change in their band gap energy. In general, the bigger size of quantum dots has the lower band gap energy, which means that they emit more red-like colors, on the other hand, the smaller quantum dots emit more violet-like colors. Because the PL spectra of the PNC@AINO was red-shifted after polymerization, we proposed that there is an increase in size of PNCs, which might be explained by slight aggregation of themselves during polymerization procedure. Similar results were observed in previous reports (*ref.* 11 and 15), and they also concluded that the red-shift behavior was caused by residual aggregation, therefore it is thought that the red-shift of PL spectra after polymerization might be a general phenomenon caused by the aggregation of PNCs, which is not just observed in or confined to this study.

Reviewer #2

Comment: The author has answered our questions very well, and the novelty and the necessity of the preparation strategies has also been highlighted and discussed in detail. It is recommended for publication as is.

Response: We appreciate the reviewer's support for publication of the revised manuscript on *Nature Communications*. Since we had additional comments from other reviewers, we have discussed those comments. It would be greatly helpful for us if the reviewer gave any comment for the discussion.

Reviewer #3

Comment: I accept the changes introduced in the publication in accordance with the comments of other reviewers. These changes made it possible to improve the publication that I find very interesting and valuable

Response: Thank you for the reviewer's support for publication of the revised manuscript on *Nature Communications*. Since we had additional comments from the other reviewer, we have discussed those comments. It would be greatly helpful for us if the reviewer gave any comment for the discussion.